# Long-term Field Experiments in Germany: Classification and spatial Representation

Meike Grosse[1], Wilfried Hierold[1], Marlen C. Ahlborn[1], Hans-Peter Piepho[2], Katharina Helming[1]

[1]Leibniz Centre for Agricultural Landscape Research (ZALF), Eberswalder Straße 84, Müncheberg, 15374, Germany
[2]Biostatistics Unit, Institute of Crop Science, University of Hohenheim, 70599 Stuttgart, Germany

*Correspondence to*: Meike Grosse (meike.grosse@zalf.de)

**Abstract.** The collective analysis of long-term field experiments (LTFE), here defined as agricultural experiments with a minumum duration of 20 years and research in the context of sustainable soil use and yield, can be used for detecting changes in soil properties and yield such as induced by climate change. However, information about existing LTFEs is
scattered, and the research data are not easily accessible. In this study, meta-information on LTFEs in Germany is compiled and their spatial representation is analysed. The study is conducted within the framework of the BonaRes project, which, inter alia, has established a central access point for LTFE information and research data. A total of 205 LTFEs is identified which fit to the definition above. Of these, 140 LTFEs are ongoing. The land use in 168 LTFEs is arable field crops, in 34 trials grassland, in two trials vegetables and in one trial pomiculture. Field crops LTFEs are categorized into fertilization
(n=158), tillage (n=38), and crop rotation (n=32; multiple nominations possible) experiments, while all grassland experiments (n=34) deal with fertilization. The spatial representation is analysed according to the climatic water balance of the growing season (1 May to 31 October) (CWBg), the Müncheberg Soil Quality Rating (MSQR) and clay content. The results show that, in general, the LTFEs well represent the area shares of both the CWBg and the MSQR classes. 89% of the arable land and 65% of the grassland in Germany is covered by the three driest CWBg classes, hosting 89% and 71% of the
arable and grassland LTFEs, respectively. LTFEs cover all six MSQR classes, however with a bias towards the high and very high soil quality classes. LTFEs on arable land are present in all clay content classes according to ESDAC, however with a bias towards the clay content class 4. Grassland LTFEs show a bias towards the clay content classes 5, 6 and 7, while well representing the other clay content classes, besides clay content class 3, where grassland LTFEs are completetly missing. The results confirm the very high potential of LTFE data for spatially differentiated analyses and modelling.
However, reuse is restricted by the difficult access to LTFE research data. The common database is an important step in overcoming this restriction.

## 1 Introduction

Long-term field experiments (LTFEs) are a valuable research infrastructure for terrestrial research in general and agricultural research in particular. They are here defined as agricultural field experiments with a minimum duration of 20 years and

30 research in the context of sustainable soil use and yield. Changes in soil properties tend to occur slowly; thus, for the identification of long-term trends, experiments with a long duration are needed. However, a single LTFE allows the drawing of conclusions only for its specific site. The collective analysis of research data from different LTFEs at different locations leads to more generalizable results. On the one hand, similar experiments on similar sites will lead to better validated conclusions when analysed in combination. On the other hand, LTFEs in different experimental conditions may lead to

35 broader implementable results by their collective analysis. Furthermore, LTFEs are expensive; a comprehensive and coordinated evaluation is also required to prove that they are worth the expense (Körschens, 2006; Berti et al., 2016). Historically, LTFEs were mainly established to answer questions regarding plant nutrition in the sense of achieving the highest possible yield (Merbach and Deubel, 2008). Later, they were used to reveal the effects of agricultural management practices (besides fertilization mainly tillage and crop rotation) on crop yield but also soil characteristics. LTFEs have been

very helpful for research on soil organic carbon content or composition (Ellerbrock and Gerke, 2016; Kaiser et al., 2014; Körschens et al., 2014). LTFEs are further important for research related to questions regarding the inter annual variability of crop yield (i.e., yield stability) that can be associated with climate change (Berti et al., 2016; Reckling et al., 2018; Macholdt et al., 2019) and respective adaptation options (Hamidov et al., 2018). Valuable data can also be delivered for the validation of models (Franko et al., 2011; Ellerbrock et al., 2005) and for concepts used to evaluate soil functions (Vogel et al., 2019;

Techen et al., 2020).

The joint analysis of LTFEs can go beyond the original research question of each LTFE, e.g., to answer questions about climate change, ecosystem services, nutrient cycles, or yield stability. This research could be done through the common assessment of the 'control' treatment of each LTFE, which is here defined as a treatment with customary tillage and fertilization and is present in most LTFEs. The combined analysis of control treatments is irrespective of the LTFE's original

research theme. This would allow us to reveal changes in soil properties independently of the original questions for which the experiments were set up, e.g., overall trends in carbon content development. Although that would be a similar analysis to what can be done with soil monitoring sites ("Bodendauerbeobachtungsflächen"), it would be a reasonable approach. It can be assumed, that LTFEs have fewer breaks during the experimental period than soil monitoring sites, as soil monitoring sites are always a "window" in real agriculture. Further on, access to data from soil monitoring sites is not neccessarily easier than

that to LTFE. Of course, the strengths of the collective analysis of LTFEs is the analysis of LTFEs with similar treatments in the form of a meta- analysis.

The meta-analyses of similar LTFE, e.g., of fertilizer experiments with similar factors (e.g., with/without organic manure) or tillage experiments (e.g., conventional tillage vs. reduced tillage) has the opportunity to make use of the original research question of the LTFE. The effects and sustainability of measures can be revealed in a broader context and in different soils.

This can be done with pairwise comparisons of alternative and reference management practices, such as that by Bai et al. (2018) and Sandén et al. (2018). However, because of the site specificity of soil-plant interactions and their responses to agricultural management practices, the upscaling and generalization of results requires information about the spatial representation of LTFE sites.

The statistical analysis of LTFEs poses several challenges and requires careful statistical modelling. We would recommend a mixed-model based analysis that accounts for the randomization layout of the trial (see Onofri et al., 2016, for review and some case studies). A general strategy starts out from the analysis model that would be used for a single year of data and then extend the model to account for variation across years. A specific challenge here is that during the course of the experiment, several observations are made on the same experimental units, and this serial correlation needs to be taken into account (Payne et al., 2015; Richter and Kroschewski, 2006; Singh und Jones, 2002). Also, there may be heterogeneity of variance between years, which may be related to changes in stability of the investigates systems (Macholdt et al. 2019a,b). For a recent account of several statistical issues in the design and analysis of LTFEs see Reckling et al. (2020).

A common issue with several LTFEs in Germany is that they were not properly randomized. This is mainly due to the fact that Fisher's principles of randomization and blocking were not widely known or accepted at the time when these trials were established. Instead, the systematic design originally proposed by Mitscherlich about a hundred years ago was very popular, and several LTFEs were established according to such systematic designs. For these unrandomized trials, a randomization-based analysis is obviously not available. One option then is to try spatial modelling, though it must be stressed that fitting of a spatial covariance structure cannot make up for lack of randomization. But such a modelling is perhaps the best way forward, if a sensible analysis is to be conducted for such trials. For a review of the connection between systematic designs as proposed by Mitscherlich and certain spatial covariance structures, see Piepho and Vo-Thanh (2020).

Important compilations of German LTFEs have been performed by Körschens (1994, 1997) and Debreczeni and Körschens (2003). In Körschens (1994), 97 German LTFEs with a duration of more than 20 years were listed. The starting year, the kind of factors, the cultivated crops, the size of the plots and experiments, the soil texture, the average annual air temperature and the average annual precipitation of the site are presented if available. In Körschens (1997), 50 German LTFEs with a duration of more than 30 years are listed, and similar information is presented. In Debreczeni and Körschens (2003), 94 German LTFEs with a duration of more than 20 years are listed, and information about the start, experimental aspects, cropping system and soil is provided. Körschens (1994, 1997) indicates the following constraints for the compilation of a complete overview of all LTFEs in Germany: the multitude of experiments, discontinued experiments, new experiments, or experiments not at all documented in the literature. In Debreczeni and Körschens (2003), restricted resources for data collection are also mentioned. In addition, the heterogeneous setup and the scattered distribution of LTFEs make comparisons of data difficult or impossible (Bai, 2018). To cope with these problems, in the frame of the project 'BonaRes', funded by the German Federal Ministry for Education and Research (BMBF), there is the focus on a central database for metadata and research data from LTFEs (BonaRes, 2020). The research data from two LTFEs (V140, Müncheberg and Dikopshof, Bonn) are available for free reuse via the BonaRes data portal (https://maps.bonares.de/mapapps/) and the research data of nine other LTFEs are very close to publication. More LTFE holders will hopefully agree to upload research data within the third (and last) funding phase of BonaRes and take the great chance for support in data processing and storage.

No information is yet available regarding the spatial representation of LTFEs in Germany with regard to important agronomic factors such as climate and soil fertility. The aim of this paper was twofold: first, to classify the LTFEs in Germany with regard to land use, research themes and farming systems. Second, the aim was to conduct a descriptive analysis of the geospatial distribution of the experimental sites with regard to key factors of agricultural production: climate and soil fertility.

## 2 Material and Methods

A combination of three methods was applied: a literature review to identify LTFEs in Germany, a fact sheet-based addition of information to the identified LTFEs, and a geospatial analysis employing the CWBg and the MSQR (Figure 1).

An extensive literature review was conducted to identify LTFEs. The search terms were 'long-term field experiment', 'long-term experiment', 'long-term field trial', and 'long-term trial', as well as the German items 'Dauerfeldversuch', 'Dauerdüngungsversuch', 'Dauerversuch', 'Langzeitfeldversuch' and 'Langzeitversuch'. Sources were scientific papers as well as other articles, books, trial guides and websites. The focus was on the exact position of the LTFE and the following metadata: name of the LTFE, website (if available), institution, land use category, participation in existing networks, research theme, size of the LTFE area, number of plots, size of the plots, crop rotation, start (and maybe end) of the trial, measured parameters, and trial setup including factors, treatments and randomization. For the coordination and simplification of the trial description, the BonaRes Fact Sheet was established, which asks for all relevant trial information (Grosse et al., 2019). It was sent to the trial holders, and the fact sheet was completed for 40 trials. Trial holders also delivered important information as personal communication.In compiling the dataset, special attention was paid to LTFEs with a minimum duration of 20 years. This age can be seen as a threshold for the identification of long-term trends. Attention was given to LTFEs in the context of soil research, i.e., the objects of research should at least include soil properties and yield as an important soil function. The setup of each trial should allow for statistical analyses, i.e., have clearly defined treatment factors, replications and as much as possible a static design. Lysimeter experiments were excluded because they were considered as an own category. Some reasons for this exclusion are that soils are often transferred and not undisturbed in lysimeter experiments and tillage has to be conducted by hand instead of machines, which can bias some results. Indeed, longterm lysimeter experiments exist in Germany as part of the TERENO network (TERENO, 2020).

The LTFEs were classified according to their research themes to simplify the identification of similar experiments. The field crops LTFEs could best be grouped into four clusters: fertilization, tillage, crop rotation, other. The fourth cluster "other" entails all themes that could not be grouped into the first three and appeared only in a few (maximum five) LTFE cases, so that a separate group was not justified. Two or more factorial experiments were sorted in all relevant classes, i.e., multiple nominations were possible. LTFEs on grassland exist only as fertilization trials.

LTFEs are precisely known in their position, and for an additional 96 LTFE the trial area is approximately known, usually on the area of the holding institution. In the latter case, either the exact position is not known or the former LTFEs are now overbuilt with streets, parking spaces or buildings.

The geospatial analysis was performed by comparing the regional distribution of LTFEs to that of (a) climatic water balance classes of the growing season (1 May to 31 October) (CWBg) and (b) the Müncheberg Soil Quality Rating (MSQR) as two complex site classifications. In addition, (c) clay content of the topsoil according to ESDAC (2020) was chosen. The representativeness of LTFEs according to the frequencies in the cells of this classification was assessed. LTFEs were classified according to their land use and their research themes to simplify the identification of similar experiments. The

identification of suitable LTFEs in similar (or different) landscapes shall be facilitated. Therefore, a table with the IDs of all experiments, their thematic classification, their CWBg class and their MSQR class is provided in the attachment. More details for each LTFE can be identified in the published dataset (Grosse and Hierold, 2019), which is freely available in the BonaRes Repository, through the ID of the LTFE. Thus, cooperation with LTFE holders can be initiated more easily.Fourteen LTFEs were excluded from the geospatial analysis because they were dealing with research themes other

than fertilization, tillage or crop rotation or did not include field crops or grassland experiments. The remaining 191 LTFEs were grouped into the four classes of fertilization experiments, tillage experiments, grassland experiments, and crop rotation experiments. The shares of LTFEs in each class were compared to that of agricultural land in Germany. For that, approximately 17.9 million hectares of agricultural land were subdivided according to their land use as arable land (approximately 13.5 million hectares) or grassland (approximately 4.4 million hectares) (Umweltbundesamt, 2019). For the

descriptive statistical analyses cross-tabulations and contingency tables were used.

The CWBg was chosen as a suitable parameter to represent the climatic conditions for agricultural land use and because of its huge relevance for vegetation growth. Its impact may be even larger than that of temperature (Crimmins et al., 2011), and it may determine the growing season (Sattar et al., 2019). We used data from the German Meteorological Service (DWD) for the period 1981-2010 for the main growing season, defined from 1 May to 31 October (Ad-hoc-AG Boden, 2005). The CWB

data for the growing season instead of the whole year was chosen, because regional differentiation is bigger for CWBg compared to the annual balance. The data are available for the whole territory of Germany with a pixel resolution of 1 km (DWD, 2020). The CWB is defined in Formula (1) as the difference in precipitation (P) and potential evapotranspiration (PET). It is a quantitative measure of the water supply in a given time period and for a specific region. The PET depends on location factors such as crop cover, topographical effects, soil conditions and soil water storage. It can therefore only be

determined selectively. However, for a better comparison for spatial calculations, the so-called grass reference evapotranspiration is considered, which indicates the evapotranspiration of a standardized grass cover in standardized soil with optimal water supply (Pereira et al., 2015).

$$CWB = P - PET \tag{1}$$

The classification of the climatic water balance in seven classes follows the Survey Guideline KA5 (Ad-hoc-AG Boden, 2005) (≤150; -150 to <-50;-50 to <50; 50 to <150; 150 to <300; 300 to <500; ≥500 mm), which are classified there from extremely low to extremely high (Ad-hoc-AG Boden, 2005).

To derive data for agricultural areas, either arable land or grassland intersections with the CORINE Land Cover (CLC, 2018) dataset were made.

For (b), a soil quality map (BGR, 2014) is used, which applies the Müncheberg Soil Quality Rating (MSQR). It has a pixel resolution of 250 m. The BGR had applied this complex assessment procedure (Mueller et al., 2010; Ad-hoc-AG Boden, 2010), which was developed as a visual procedure for estimating yield potential in the field, by modelling data from the soil overview map (BGR, 2007), but only for arable land. It takes soil structure and soil degradation threats into account and integrates eight basic soil indicators with 13 hazard indicators into a rating of soil quality. The rating is shown on an ordinal scale of 0 to 102 and clustered into six quality classes, with higher values indicating higher yield potential (Daedlow, 2018). The eight soil indicators are substrate, A-horizon depth, topsoil structure, subsoil structure, rooting depth, profile available water, wetness and ponding, slope, and relief. The 13 hazard indicators are contamination, salinization, sodification, acidification, low total nutrient status, shallow soil depth above hard rock, drought, flooding and extreme waterlogging, steep slope, rock and surface, high percentage of coarse texture fragments, a soil thermal regime unsuitable for crop production, and miscellaneous hazards (e.g., exposure to wind and water erosion). Most of the indicators are sensitive to agricultural management, which makes the MSQR most useful for studying the effects of agricultural management on soil. The MSQR has been proven useful in other studies of geo-spatial representation (Askari et al., 2013; Hanauer et al., 2017; Smolentseva et al., 2014). Since no MSQR is available for grassland areas, the LTFEs on grassland were excluded in this analysis.

Out of the 157 fertilization, tillage or crop rotation LTFEs on arable land, 26 could not be assigned to a class of MSQR because the fields are surrounded by buildings and are therefore not part of arable land. If an LTFE did not obtain an assignment at a GIS intersection, the value was determined manually by plausibility examination of the nearest 5 to 7 grid cells. One LTFE could not be assigned to a class of MSQR because it compares three different soils in boxes.

For (c), clay content, data of the European Soil Data Centre (ESDAC) based on LUCAS topsoil data is used (ESDAC, 2020). Although clay content is included in the MSQR as part of substrate, we decided to analyse the area shares of clay content separately, as carbon content is often correlated with the clay content (Körschens, 1997). Moreover, clay content is needed to estimate the carbon balance in a model derived from the CANDY model (Franko et al., 2011). Further on, ESDAC offers international data, therefore clay content is suitable for international comparability. Due to the fact, that texture is part of the MSQR, we do not offer separate maps for clay content, but present data in tables.

Calculations always refer to utilized agricultural areas or parts thereof, arable land or grassland.

The information was analysed with Microsoft Excel. The geospatial analysis was performed using the ESRI software ArcMap 10.6.1 (ESRI, 2018).

The research on LTFEs is not completed but is ongoing. The information about LTFEs is continuously updated and expanded. New LTFEs are integrated, and the information about each LTFE is extended. The state of research is November 2019.

## 3 Results and Discussion

### 3.1 Overview of LTFEs in Germany

In total, 205 LTFEs across Germany with a minimum duration of 20 years were identified, of which 140 trials are ongoing and 65 are terminated (status: November 2019). Further LTFEs reaching the 20-year threshold within the next five years (until 2024) were also included (n=6; Figure 2). Most of the trials have a duration between 20 and 49 years (n=124; Figure 2). 50 trials have a duration between 50 and 99 years. Three trials have been running for more than 100 years ('Ewiger

Roggen', Halle, 1878 - today; 'Statischer Düngungsversuch V120', Bad Lauchstaedt, 1902 – today; 'Dauerdüngungsversuch Dikopshof', Wesseling, 1904 - 2009). The age of 22 terminated trials is unknown since only the starting date of the trials is known but not the exact ending year. As these trials were mentioned in different important sources as being ongoing (Amberger and Gutser, 1976; Debreczeni and Körschens, 2003; Körschens, 1990, 1994, 1997, 2000), it is known that their duration was at least 20 years.

The land use in 168 LTFEs is arable field crops, in 34 trials grassland, in two trials vegetables and in one trial pomiculture . There are more long-term grassland experiments in Germany; we have not included them in our research because they are dedicated to research themes other than questions of sustainable soil use and yield.

The majority of LTFEs were established after 1947, when research was resumed after the Second World War (Figure 3). In 1996/1997, a series of grassland fertilization experiments was established by several German state authorities. This explains

the high number of LTFEs established in these years (Figure 3).

The research themes of the LTFEs can be assigned to the following categories: fertilization, tillage, crop rotation, 'other' themes and combinations of these (Table 1). Due to trials with two or more treatment factors, multiple nominations of experiments for the different research themes were assigned (n=251). Most LTFEs were established for research on fertilization (Figure 3 and Table 1) (n=158). This result is coincident with the results from a study in the international context

(Berti et al., 2016). In Germany, fertilization LTFEs can be subdivided into field crop experiments (n=124) and grassland experiments (n=34). Historically, questions regarding the effects of fertilization on plant growth were the focus of research, while more recently the effects on the soil and the environment are investigated. In the focus of the experiments are either different kinds of fertilizers or different amounts of fertilizers or comparisons with/without a specific fertilizer or combinations of these. Most frequently, organic fertilization versus mineral N fertilization is examined. In fewer

experiments, the effect of straw fertilization is the subject of research. Additionally, the effects of mineral K fertilization, mineral P fertilization, liming, green manure, mineral Mg fertilization, compost, or sludge are examined (Table 1). More rarely, different points in time of the fertilizing measure are compared.

Thirty-eight LTFEs address tillage variations (Table 1). Most of these tillage experiments compare different tillage intensities. Most often, reduced tillage depth or conservation tillage are the subjects of research. Also, inversion versus non-inversion tillage is compared. Further research themes are sowing methods, different forms of primary tillage, the effects of stubble tillage, and tillage frequency (Table 1). The oldest tillage experiment started in 1923 (Statischer Dauerversuch Bodennutzung, Berlin-Dahlem), but 25 tillage experiments started in 1990 or later (Figure 3). Therefore, most of the tillage experiments are 'younger' experiments, a result also congruent with the findings of Berti et al. (2016).

Thirty-two LTFEs have the research theme 'crop rotation'. Mostly, the effect of crop rotation on soil properties and yield is investigated. Therefore, rotational cropping versus monoculture is compared. Additionally, plant health is the focus, e.g., compatibility of different cereal species or different percentages of cereals in crop rotation (Table 1). Most of the crop rotation experiments were established after 1950. 19 experiments of the 32 crop rotation experiments are still ongoing. The oldest crop rotation experiment, the 'Eternal Rye', was established in 1878 by the Martin Luther University of Halle.

Twenty-three trials address research themes other than fertilization, tillage or crop rotation. The 'other' research themes are highly diverse. 'Environmentally friendly crop protection', mainly reduced pesticide intensity, is the most frequent research theme among the 'other' research themes (n=5). 'Irrigation' is the second most frequent (n=4). 'Effects of different forms of fallow' is within the focus of three LTFEs. 'Frequency and start of utilization of grassland', 'Land use systems comparison', 'Monitoring of Organic Farming' and 'Use of biodynamic preparations' are each within the focus of two LTFEs. Three other research themes are present in only one LTFE (Table 1).

Many different parameters are measured in LTFEs. In Grosse et al. (2019) 46 different soil parameters and 29 plant parameters are listed, which were measured in LTFEs. The analysed parameters can be assigned to different soil functions. The following five soil functions were chosen as most relevant for BonaRes: biomass production, water storage and filtering, nutrient storage and recycling, carbon storage, and habitat for biological activity. In most LTFEs, parameters for biomass production were measured like yield and yield components. Nutrient storage and recycling is the second frequent soil function. Less research is conducted (in decreasing frequency) for carbon storage, habitat for biologic activity and water storage and filtering.

Archived samples are an important means of performing or repeating measurements. However, the information, if archived samples exist, is difficult to find in the literature. We have the information from a fact sheet query. Of 40 responses received, 32 LTFEs have archived samples. A total of 184 trials are set up with conventional management practices, 14 with organic management practices and five with so-called integrated agriculture. Two trials compare conventional with organic management practices.

The holding institution for 96 trials is a university or university of applied sciences, and for 61 trials, it is a state authority. 27 trials are in the responsibility of non-university scientific institutions such as research institutes. 21 trials are or were held by industry.

Compared to LTFEs worldwide, there are a comparatively large number of LTFEs in Germany. Our research revealed up to now 177 LTFEs which match our definition in the following countries: Austria, Belarus, Belgium, Bulgaria, China, Czech

Republik, Denmark, Estonia, Finnland, France, Hungary, Ireland, Italy, Moldova, Norway, Poland, Romania, Russia, Slovenia, Spain, Sweden, Switzerland, United Kingdom, Ukraine, and USA. They are comparable in age (the oldest ones started 1843) and research themes. There are international networks such as the working group IOSDV (Internationale Organische Stickstoffdauerdüngungsversuche, Körschens, 2000), the GLTEN (Global Long-Term Experiment Network, GLTEN, 2020), which was launched in 2018, and networks of organic LTFEs like RetiBio in Italy and RotAB network in France (Ciaccia et al., 2020). In order to make the best use of the great efforts and costs that are behind every single LTFE, international networks should cooperate more intensively in future and possibly also use data infrastructures jointly. We would like to point out that the BonaRes data repository can also be used by international data holders.

All information about the LTFEs in Germany is published in an online overview map (https://ltfe-map.bonares.de). The aims of the overview map are to make LTFEs more visible, to enhance networking among LTFEs and to simplify joint analyses of LTFEs. It is available in German and English. The map content can be displayed according to different categories, e.g., the research themes, land use, or duration of the LTFEs. In addition to the overview information, details about every single LTFE are provided in a pop-up window, offering valuable information for potential users for orientation and initiation of cooperation.

As limitations of existing LTFEs it can be mentioned, that erosion and compaction are typically not analysed in LTFEs and they are not designed for such questions up to now. Grassland experiments are in fact meadow experiments, whereas grazing experiments are completely missing.

### 3.2 Geospatial Analyses

### 3.2.1 Geospatial Analysis of LTFEs in Relation to the Climatic Water Balance of the growing season (CWBg) Distribution

An overview of the distribution of these CWBg classes and of LTFEs in Germany is given in Figure 4. According to Table 2 and Figure 4, arable land is distributed among classes 1-7 of the CWBg (Table 2; Figure 4): the largest shares of 33% each are classified as CWBg classes 2 (from -150 mm to <-50 mm) or 3 (from -50 mm to <50 mm), respectively. The area of CWBg class 2 is mainly located in the lowlands of Germany: in the western and northern Rhine-Main Valley, in a majority of the north-eastern lowland and the Loess Boerde. The area of CWBg class 3 is mainly distributed in the north-eastern part of Germany and in parts of the Southern German Escarpment Landscape, the northern foothills of the Alps (lower Bavarian upland) and the lower uplands, such as the Lower Saxon and Hessian lowlands, the Vogtland district and the Erzgebirge foreland. 23% of the arable land is allotted to CWBg class 1 (<-150 mm). This extremely low CWBg is located almost exclusively in eastern Germany, especially in the rain shadow of the Harz: the Fläming, the plates and lowlands of mid Brandenburg and the heathland of Brandenburg. Minor shares of 7% and 4% are allotted to CWBg classes 4 (from 50 mm to <150 mm) and 5 (from 150 mm to <300 mm), respectively. CWBg class 4 is located mainly in the foothills of the Alps and around the secondary mountains and in the western Schleswig-Holstein (moraines of Schleswig-Holstein). CWBg 5 is

mainly located in Germany's southern foothills of the Alps. CWBg class 6 (from 300 mm to <500 mm) is not present in Germany's arable land, and CWBg class 7 (>500 mm) is not present in Germany's agricultural land (arable and grassland).

Among the grassland, the largest share of 33% is classified as CWBg class 3 (Table 3). 23% of grassland are classified as CWBg class 5. 18% are classified as CWBg class 2, 14% as CWBg class 1 and 9% as CWBg class 4. CWBg class 6 is present in a small share (3%) of Germany's grassland at higher altitudes in the Alpine region.

To analyse sites in every CWBg class, each class would have to be represented through LTFEs. Ideally, the shares of LTFEs in each class would correspond to the agricultural area. This is, of course, not the case (Table 2), as LTFEs were not established systematically in the landscape. Each CWBg class present in the arable land is represented by LTFEs, but they are not found in the same shares. CWBg class 1 is overrepresented by all LTFE types, CWBg class 2 is underrepresented by crop rotation LTFEs, class 3 is underrepresented by fertilization LTFEs and crop rotation LTFEs, class 4 is underrepresented by tillage LTFEs and overrepresented by crop rotation LTFEs (although in number, there are only 4 crop rotation LTFEs), and class 5 again is overrepresented by crop rotation LTFEs (although in number, there are only 6 crop rotation LTFEs) (Table 2; Figure 4) Overall, the three CWBg classes 1-3 representing 89% of the arable land area also host 89% of the LTFEs with a certain bias towards the driest CWBg class 1. Given that no spatial planning was considered during the allocation of LTFEs, this is a remarkably good distribution.

Among grassland LTFEs, not every CWBg class is represented by LTFEs (Table 3). Thus, CWBg class 6 is present in a small share of grassland (3%) but is not represented by any grassland LTFEs. CWBg classes 2 and 5 are underrepresented by grassland LTFEs, while CWBg classes 3 and 4 are overrepresented by grassland LTFEs. Overall and compared to the arable land area, the three driest CWBg classes 1-3 represent only 65% of the grassland area and host 71% of the grassland LTFEs.

### 3.2.2 Geospatial Analysis of LTFEs in Relation to the Müncheberg Soil Quality Rating (MSQR) Distribution

An overview of the distribution of the MSQR classes and of LTFEs in Germany is given in Figure 5. Soils classified as 'very high' are located mainly in the central part of Germany. Soils classified as 'high' exist in the central part and in the south of Germany as well as in some smaller areas in the north-western region of Germany, including the coastlines. Soils classified as 'low' and 'medium' are predominant in the northern part of Germany but also exist in some areas in the middle and south of Germany. Soils classified as 'very low' mainly exist in north-eastern Germany. Soils classified as 'extremely low' exist mainly in small areas of mid-east and mid-west and north-west Germany (Figure 5).

The classification of the agricultural area into the six MSQR classes (Table 4) is as follows: The largest share (28%) of agricultural area is classified as 'medium'. The smallest shares are classified as 'extremely low' (6%) and 'very high' (10%). Medium shares are classified as 'very low' (17%), 'low' (21%) and 'high' (18%). LTFE sites exist in all MSQR classes, and overall, the distribution of the LTFE sites follows a similar pattern as that of the MSQR classes, with the exception of a bias towards the 'high' MSQR class.

### 3.2.3 Geospatial Analysis of LTFEs in Relation to the combined CWBg and MSQR Distribution

The share of the arable area in Germany and the share of LTFEs on arable land in every CWBg-MSQR intersection are compared (Figure 6). According to this analysis, in the MSQR class 'extremely low', the share of LTFEs matches the share of arable land area in each CWBg class. In the other MSQR classes, CWBg 1 is overrepresented by LTFEs compared to the respective land area. Thus, regarding climate, the distribution of LTFEs is biased towards dry areas with very low CWBg class 1. The reason for this bias is probably because most of these LTFEs are located in the region surrounding Berlin and the region Bad Lauchstädt/Halle/Seehausen, which are both historical agricultural research areas.

In CWBg class 2, the distribution of LTFEs is biased towards high and very high MSQR classes. This result is mainly caused by the sites Bonn, Braunschweig, Gießen and Göttingen.

CWBg class 3 is underrepresented by LTFEs in the MSQR classes of very low, low, medium and high.

CWBg classes 4 and 5 are rather adequately represented by LTFEs in every MSQR class. However, these CWBg classes rarely exist in Germany.

For the landscape approach proposed in this paper, more LTFEs would be required in areas with CWBg class 3 on soils classified as MSQR 'very low', 'low', 'medium' and 'high' and in areas with CWBg class 2 on soils classified as MSQR 'very low', 'low' and 'medium'.

### 3.2.4 Geospatial Analysis of LTFE in Relation to the clay content Distribution

According to Table 5, every clay content class is represented by LTFEs on arable land. Clay content class 4 (17% to 19% clay content) is overrepresented by LTFE, while the high clay content classes 7 (25% to 27% clay content) and 8 (28% to 98% clay content) are underrepresented, especially by fertilization and crop rotation LTFEs.

Among grassland, LTFEs in clay content class 3 (11% to 16% clay content) are completely missing (Table 6). The clay content classes 5 (20% to 21% clay content), 6 (22% to 24% clay content) and 7 (25% to 27% clay content) are overrepresented by grassland LTFEs, while the other clay content classes are rather equally represented.

Franko et al. (2011) found in their analysis of 40 LTFEs for the validation of a C-Model that more experimental results on clay soils would be required. This could be confirmed for LTFEs on arable land in this study.

### 4 Conclusions

To obtain adequate information about each CWBg, MSQR and clay content class through LTFEs, more LTFEs would have to be established. However, nearly every class is represented by at least some LTFEs. For the joint analysis, there are other, more important constraints: data are not easy to access, and sometimes the older data are not digitized. Here, BonaRes offers great opportunities through the provision of support for data preparation and through the establishment of a common database. We hope that this great opportunity will be frequently used by LFTE holders in future.

## Data availability

The LTFE metadata are available in the BonaRes Respository: Grosse, M., and Hierold, W.: Long-term Field Experiments in Germany [Data set], BonaRes, http://doi.org/10.20387/BonaRes-3tr6-mg8r, 2019.

## Author Contribution

Conceptualization: Meike Grosse, Katharina Helming, Wilfried Hierold

Data curation: Meike Grosse and Wilfried Hierold

Formal analysis: Meike Grosse, Marlen C. Ahlborn, Hans-Peter Piepho

Supervision: Katharina Helming

Visualization: Meike Grosse, Marlen C. Ahlborn

Writing: Meike Grosse with contributions from all authors

## Acknowledgements

This work was funded by the German Federal Ministry of Education and Research (BMBF) in the framework of the funding measure 'Soil as a Sustainable Resource for the Bioeconomy – BonaRes', project 'BonaRes (Module B): BonaRes Centre for Soil Research, subproject B' (Grants 031A608B and 031B0511B). Many thanks to Biljana Savic and Kevin Urbasch for help with GIS-analyses. Many thanks to three anonymous referees for most valuable and in-depths comments.

## Declaration of Interest Statement

The authors declare that they have no conflict of interest.

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

**Figures and Tables**

(In the order of their appearance)

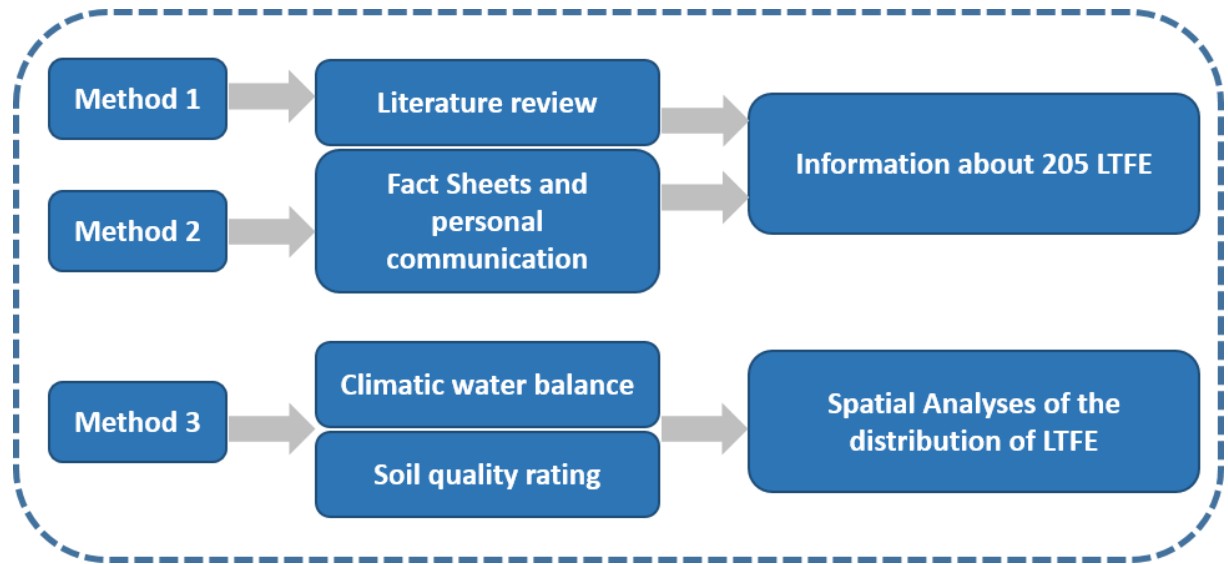

Figure 1: Methods used for assessing the representativeness of the LTFE distribution in Germany.

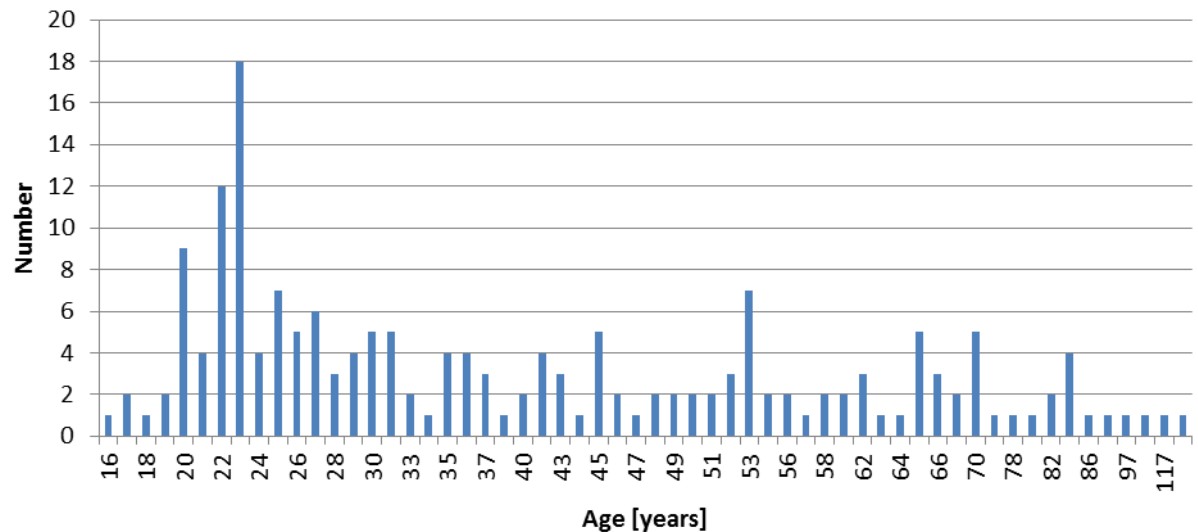

Figure 2: Number of LTFEs per age in 2019 (n=183; age of 22 LTFEs unknown)

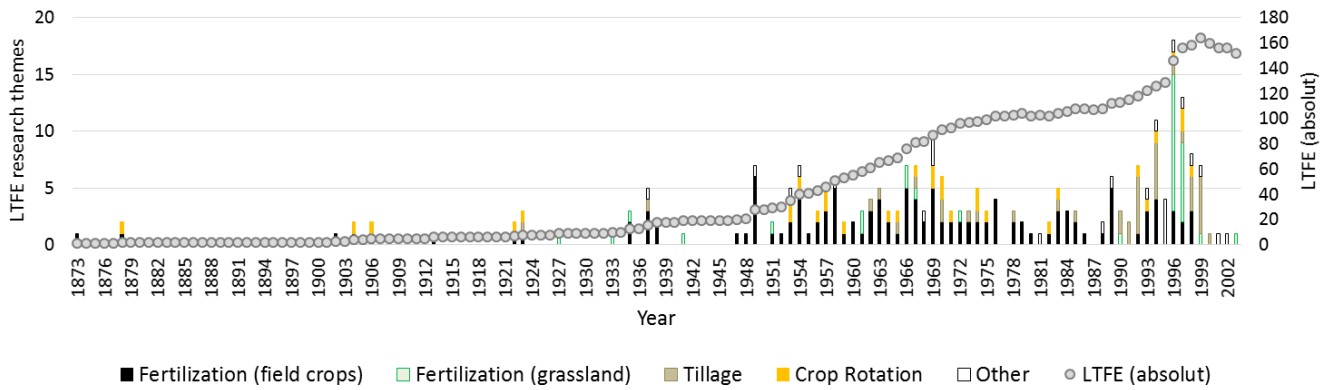

Figure 3: Number of LTFEs' set up per year according to the research themes of the experiments (multiple nominations possible, n=251) and total number of LTFEs per year (= established LTFE minus terminated LTFE).

Table 1: Research themes in LTFEs (multiple nominations possible, sorted by frequency).

| Theme | Number of trials |
|---|---|
| **Fertilization – field crops experiments** | **124** |
| Manure fertilization | 58 |
| Mineral N-fertilization | 55 |
| Straw fertilization | 24 |
| Mineral K-fertilization | 15 |
| Mineral P-fertilization | 14 |
| Liming | 10 |
| Green manure (with *vs*. without) | 8 |
| Mineral fertilization (not specified) | 6 |
| Mineral Mg-fertilization | 4 |
| Compost | 3 |
| Sludge | 2 |
| **Tillage – field crops experiments** | **38** |
| Reduced depth or conservation tillage | 24 |
| Inversion *vs*. non-inversion tillage | 12 |
| Sowing methods | 10 |
| Different forms of primary tillage | 7 |
| Stubble tillage (with *vs*. without) | 3 |
| Tillage frequency | 3 |

| | |
|---|---|
| Other | 2 |
| **Fertilization – grassland experiments** | **34** |
| Mineral P-fertilization | 11 |
| Mineral K-fertilization | 10 |
| Mineral N-fertilization | 6 |
| Liming | 4 |
| Manure fertilization | 2 |
| Sludge | 2 |
| Mineral fertilization (not specified) | 1 |
| Acid *vs*. alkaline fertilization | 1 |
| **Crop rotation – field crops experiments** | **32** |
| Crop rotation (not specified) | 23 |
| Rotational cropping *vs*. monoculture | 4 |
| Effect of pre crop | 2 |
| Crop rotation organic *vs.* integrated | 1 |
| Different percentages of cereals | 1 |
| Different percentages of wheat | 1 |
| **Other – field crops and grassland experiments** | **23** |
| Crop protection | 5 |
| Irrigation | 4 |
| Effects of different forms of fallow | 3 |
| Frequency and start of utilization of grassland | 2 |
| Land use systems comparison | 2 |
| Monitoring of Organic Farming | 2 |
| Use of biodynamic preparations | 2 |
| Chopped woody plants for weed suppression | 1 |
| Effect of weather conditions | 1 |
| Thistle control | 1 |

Table 2: Climatic water balance of the growing season (1 May to 31 October) (CWBg) classification of arable land in Germany and the number or share of the different LTFE types in each CWBg class.

| CWBg class | Range [mm/yr] | | Agricultural area (arable) | | LTFE total (arable land) (n=169) | | Fertilization LTFE* (n=124) | | Tillage LTFE* (n=38) | | Crop rotation LTFE* (n=32) | |
|---|---|---|---|---|---|---|---|---|---|---|---|---|
| | | | area [ha] | share [%] | number | share [%] | number | share [%] | number | share [%] | number | share [%] |
| 1 | | <-150 | 3 135 676 | 23 | 66 | 39 | 49 | 40 | 13 | 34 | 13 | 41 |
| 2 | -150 - | <-50 | 4 473 111 | 33 | 49 | 29 | 39 | 31 | 12 | 32 | 6 | 19 |
| 3 | -50 - | <50 | 4 468 852 | 33 | 35 | 21 | 21 | 17 | 11 | 29 | 3 | 9 |
| 4 | 50 - | <150 | 926 798 | 7 | 10 | 6 | 10 | 8 | 1 | 3 | 4 | 13 |
| 5 | 150 - | <300 | 492 110 | 4 | 9 | 5 | 5 | 4 | 1 | 3 | 6 | 19 |
| 6 | 300 - | <500 | 0 | 0 | 0 | 0 | 0 | 0 | 0 | 0 | 0 | 0 |
| 7 | | >500 | 0 | 0 | 0 | 0 | 0 | 0 | 0 | 0 | 0 | 0 |

*multiple nominations possible

Table 3: Climatic water balance of the growing season (CWBg) classification of agricultural area used for grassland in Germany and the number or share of the LTFEs on grassland in each CWBg class.

| CWBg class | Range [mm/yr] | | Agricultural area (grassland) | | Grassland LTFE (n=34) | |
|---|---|---|---|---|---|---|
| | | | area [ha] | share [%] | number | share [%] |
| 1 | | <-150 | 599 247 | 14 | 6 | 18 |
| 2 | -150 - | <-50 | 792 064 | 18 | 3 | 9 |
| 3 | -50 - | <50 | 1 420 319 | 33 | 15 | 44 |
| 4 | 50 - | <150 | 398 496 | 9 | 7 | 21 |
| 5 | 150 - | <300 | 1 009 952 | 23 | 3 | 9 |
| 6 | 300 - | <500 | 137 968 | 3 | 0 | 0 |
| 7 | | >500 | 0 | 0 | 0 | 0 |

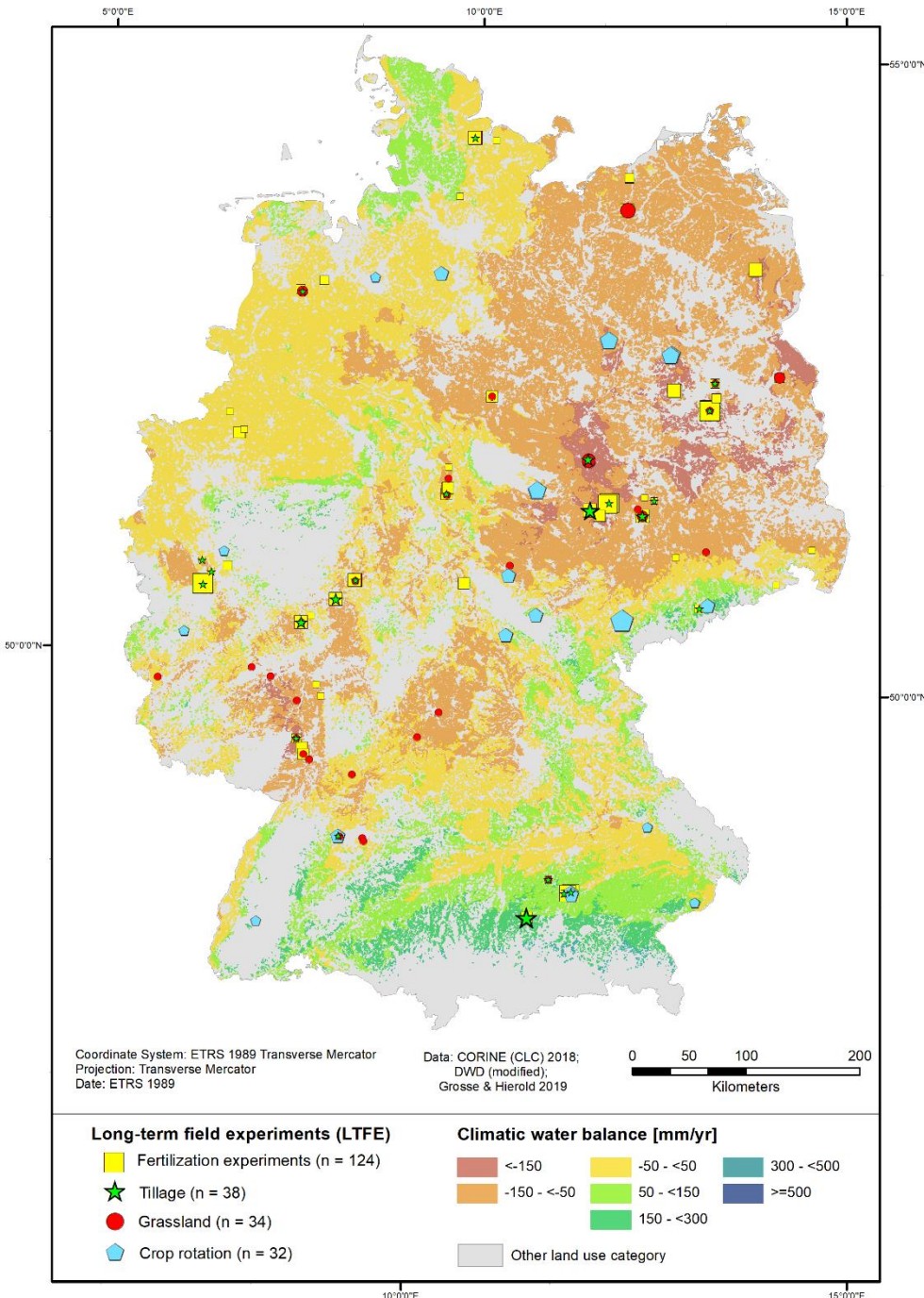

Figure 4: Overview of the distribution of the different climatic water balance classes of the growing season and the different LTFE types in Germany.. The size of the symbols varies according to the amount of LTFEs at one place.

Table 4: Müncheberg Soil Quality Rating (MSQR) classification of arable land in Germany and the number or share of the different LTFE types in each MSQR class.

| MSQR | Agricultural area | | LTFEs total (arable land) (n=169) | | Fertilization LTFEs* (n=123) | | Tillage LTFEs* (n=38) | | Crop rotation LTFEs* (n=32) | |
|---|---|---|---|---|---|---|---|---|---|---|
| | area [ha] | share [%] | number | share [%] | number | share [%] | number | share [%] | number | share [%] |
| extremely low | 705 687 | 6 | 9 | 5 | 5 | 4 | 4 | 11 | 3 | 9 |
| very low | 2 149 584 | 17 | 29 | 17 | 22 | 18 | 5 | 13 | 5 | 16 |
| low | 2 656 535 | 21 | 18 | 11 | 13 | 11 | 3 | 8 | 1 | 3 |
| medium | 3 532 109 | 28 | 32 | 19 | 28 | 23 | 6 | 16 | 4 | 13 |
| high | 2 182 221 | 18 | 45 | 27 | 28 | 23 | 13 | 34 | 11 | 34 |
| very high | 1 181 237 | 10 | 36 | 21 | 27 | 22 | 7 | 18 | 8 | 25 |

*multiple nominations possible

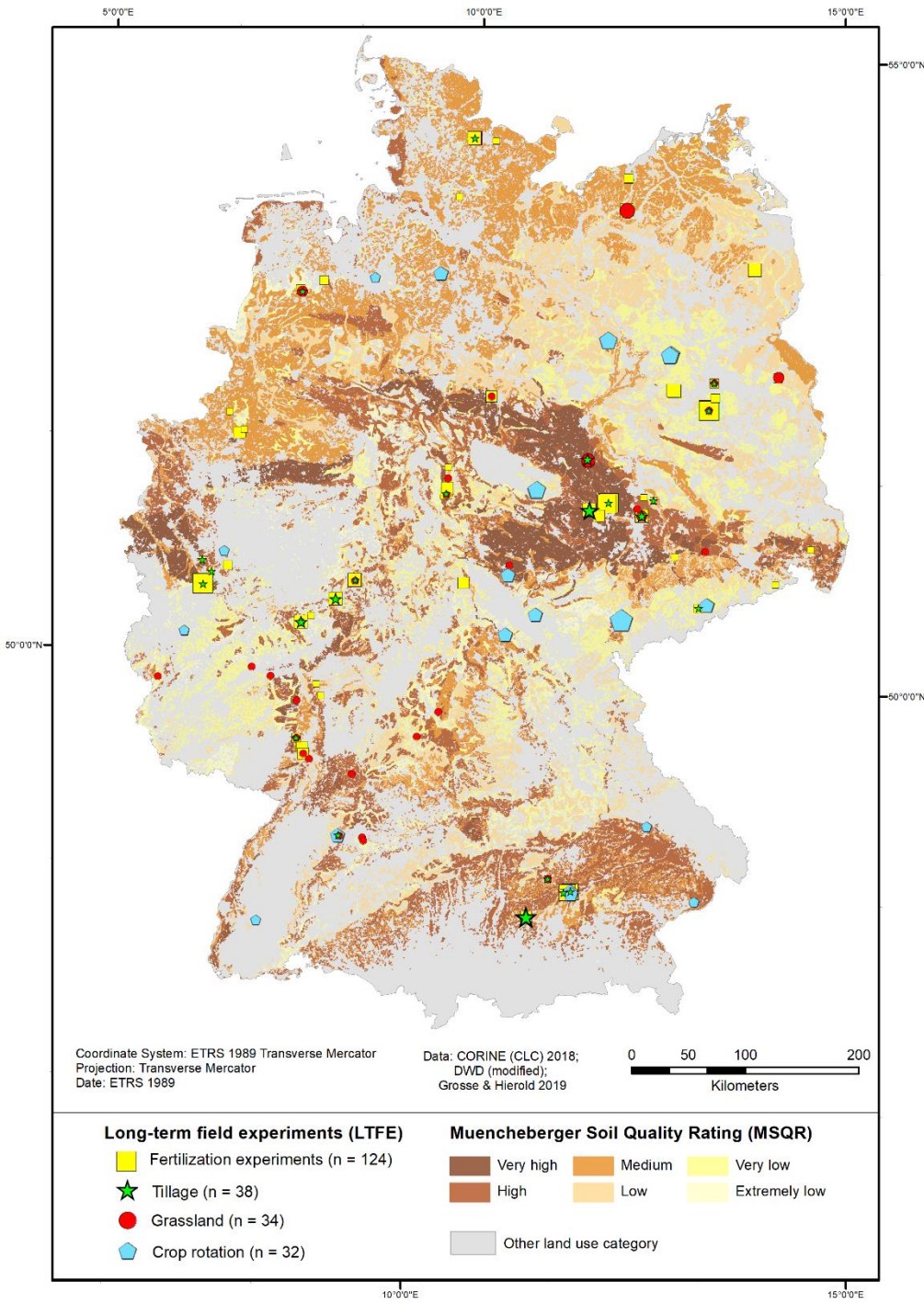

Figure 5: Overview of the distribution of the different Müncheberg Soil Quality Rating classes and the different LTFE types
in Germany. The size of the symbols varies according to the amount of LTFEs at one place.

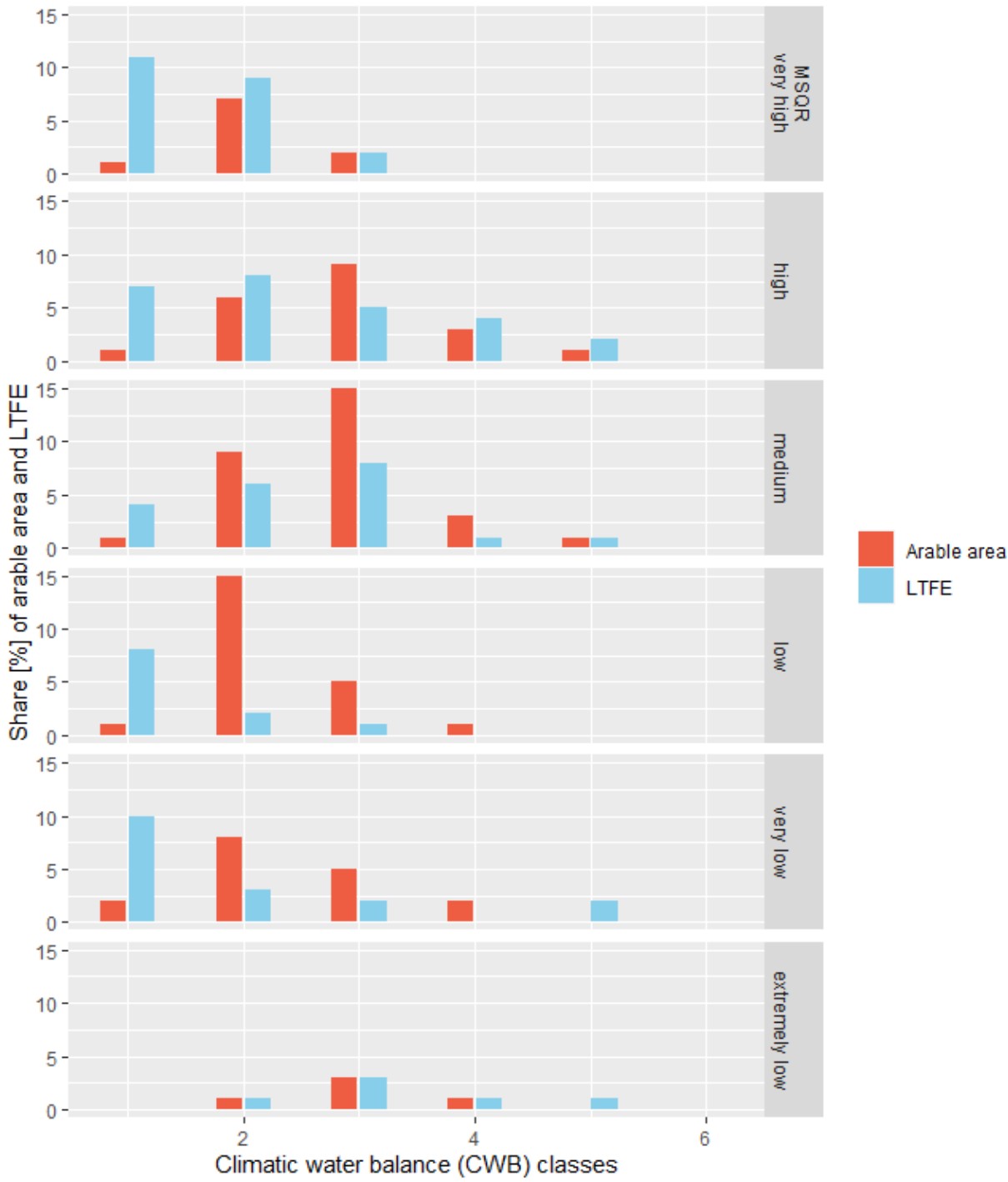

Figure 6: Share of arable area and LTFEs in every climatic water balance – Müncheberg Soil Quality Rating combination.

Table 5: Clay content classification according to ESDAC (2020) of arable land in Germany and the number or share of the different LTFE types in each clay content class.

| Clay content class | Range [%] | Agricultural area (arable) | | LTFEs total (arable land) (n=169) | | Fertilization LTFEs* (n=124) | | Tillage LTFEs* (n=38) | | Crop rotation LTFEs* (n=32) | |
|---|---|---|---|---|---|---|---|---|---|---|---|
| | | area [ha] | share [%] | number | share [%] | number | share [%] | number | share [%] | number | share [%] |
| 1 | 0 to 5 | 1 748 393 | 14 | 25 | 15 | 19 | 15 | 6 | 16 | 3 | 9 |
| 2 | 6 to 10 | 2 404 798 | 19 | 24 | 14 | 19 | 15 | 6 | 16 | 3 | 9 |
| 3 | 11 to 16 | 2 265 517 | 18 | 29 | 17 | 20 | 16 | 5 | 13 | 4 | 13 |
| 4 | 17 to 19 | 1 523 493 | 12 | 42 | 25 | 37 | 30 | 6 | 16 | 6 | 19 |
| 5 | 20 to 21 | 1 179 602 | 9 | 15 | 9 | 12 | 10 | 2 | 5 | 8 | 25 |
| 6 | 22 to 24 | 1 553 463 | 12 | 20 | 12 | 11 | 9 | 5 | 13 | 6 | 19 |
| 7 | 25 to 27 | 1 097 725 | 9 | 4 | 2 | 1 | 1 | 3 | 8 | 1 | 3 |
| 8 | 28 to 98 | 1 082 066 | 8 | 10 | 6 | 5 | 4 | 5 | 13 | 1 | 3 |

Table 6: Clay content classification according to ESDAC (2020) of agricultural area used for grassland in Germany and the number or share of the LTFEs on grassland in each clay content class.

| Clay content class | Range [%] | Agricultural area (grassland) | | Grassland LTFEs (n=34) | |
|---|---|---|---|---|---|
| | | area [ha] | share [%] | number | share [%] |
| 1 | 0 to 5 | 715 137 | 11 | 3 | 9 |
| 2 | 6 to 10 | 941 166 | 15 | 5 | 15 |
| 3 | 11 to 16 | 952 126 | 15 | 0 | 0 |
| 4 | 17 to 19 | 821 432 | 13 | 4 | 12 |
| 5 | 20 to 21 | 710 826 | 11 | 6 | 18 |
| 6 | 22 to 24 | 978 366 | 15 | 5 | 15 |
| 7 | 25 to 27 | 651 066 | 10 | 8 | 24 |
| 8 | 28 to 98 | 639 561 | 10 | 3 | 9 |

Table A 1: IDs of all long-term field experiments, their original name, their place, their CWBg class (1 May to 31 October), their MSQR class, and their thematic classification. The institutional address is indicated by a number and given below the table. More details about the LTFEs can be found in the complete dataset (Grosse & Hierold, 2019).

| ID | LTFE Name | Place of LTFE | Address (see below) | CWBg Class | MSQR Class | Thematic Classification |
|----|-----------|---------------|---------------------|------------|------------|-------------------------|
| | | **Fieldcrops LTFE** | | | | |
| 1 | Bodenbearbeitungsversuch Dichtelbach | Dichtelbach (Hunsrück) | 1 | 3 | very low | Tillage |
| 2 | Bodenbearbeitungsversuch Welschbillig | Welschbillig (Eifel) | 1 | 3 | very low | Tillage |
| 3 | Bodenbearbeitungsversuch Wintersheim | Wintersheim (Rheinhessen) | 1 | 1 | very high | Tillage |
| 4 | Statischer Düngungsversuch V120 | Bad Lauchstädt | 2 | 1 | very high | Fertilization |
| 5 | Erweiterter Statischer Düngungsversuch V120a | Bad Lauchstädt | 2 | 1 | very high | Fertilization |
| 6 | Modellversuch Stalldungsteigerung | Bad Lauchstädt | 2 | 1 | very high | Fertilization |
| 7 | Bracheversuch V505a | Bad Lauchstädt | 2 | 1 | very high | Other |
| 8 | Statischer Stickstoffdüngungsversuch | Bad Salzungen | 3 | 2 | very low | Fertilization |
| 9 | Statischer Kalkdüngungsversuch M16 | Bad Salzungen | 3 | 2 | very low | Fertilization |
| 11 | Dauerdüngungsversuch L28 | Bad Salzungen | 3 | 2 | very low | Fertilization |
| 13 | Statischer Dauerversuch Bodennutzung (BDa_D3) | Berlin-Dahlem | 4 | 1 | very low | Fertilization/Tillage/ Crop rotation |
| 14 | Internationaler Organischer- | Berlin-Dahlem | 4 | 1 | very low | Fertilization |

| | | | | | | |
|---|---|---|---|---|---|---|
| | Stickstoff-Dauerdüngungsversuch (BDa_IOSDV) | | | | | |
| 15 | Agrarmeteorologisches Intensivmessfeld (BDa_E-Feld) | Berlin-Dahlem | 4 | 1 | very low | Other |
| 16 | Bodenbearbeitungsversuch (Versuchsfeld Westerfeld) | Bernburg-Strenzfeld | 5 | 1 | very high | Tillage |
| 17 | Anbausysteme-Vergleich | Bernburg-Strenzfeld | 6 | 1 | very high | Crop rotation/Other |
| 18 | Grundbodenbearbeitung und Distelbekämpfung, ö• kologisch viehlos | Bernburg-Strenzfeld | 6 | 1 | very high | Tillage/Crop rotation/Other |
| 19 | Bodenbearbeitung und Bestelltechnik in der Fruchtfolge | Bernburg-Strenzfeld | 6 | 1 | very high | Tillage/Other |
| 20 | Dauerdüngungsversuch Dikopshof | Wesseling-Dikopshof | 7 | 2 | very high | Fertilization/Crop rotation |
| 21 | Selektions-Dauerversuch SDV | Klein Altendorf | 7 | 3 | very high | Crop rotation |
| 22 | Strohdüngung zu Getreide | Meckenheim | 7 | 2 | very high | Fertilization |
| 23 | Phosphatformenversuch | Meckenheim | 7 | 2 | very high | Fertilization |
| 24 | Organische Düngung | Meckenheim | 7 | 2 | very high | Fertilization |
| 25 | Strohdüngung mit Faulschlamm | Meckenheim | 7 | 2 | very high | Fertilization |
| 26 | Kaliformenversuch | Meckenheim | 7 | 2 | very high | Fertilization |
| 27 | Strohdüngung mit verschiedenen N-Formen | Meckenheim | 7 | 2 | very high | Fertilization |
| 28 | Phosphatvorratsdüngung | Meckenheim | 7 | 2 | very high | Fertilization |
| 29 | Kalkversuch mit Spurenelementen | Meckenheim | 7 | 2 | very high | Fertilization |
| 30 | Versuch mit Faulschlämmen | Meckenheim | 7 | 2 | very high | Fertilization |

| 31 | Dauerdüngungsversuch | Bonn-Poppelsdorf | 7 | 2 | high | Fertilization/Crop rotation |
| 33 | Langzeit Düngungsversuch (FV4) | Völkenrode | 8 | 2 | very high | Fertilization/Tillage |
| 34 | C-Dauerfeldversuch (FV36) | Völkenrode | 8 | 2 | very high | Fertilization |
| 35 | Südfeld-Düngungsversuch | Völkenrode | 9 | 2 | very high | Fertilization |
| 36 | Folgenabschätzung der Wechselwirkung von Fruchtfolge, Düngung und Pflanzenschutz | Dahnsdorf | 10 | 1 | high | Other |
| 37 | Langzeit-Düngungsversuch | Darmstadt | 11 | 2 | low | Fertilization |
| 38 | Klassischer DFV (4b2, organische und mineralische Düngung) | Dülmen | 12 | 3 | medium | Fertilization |
| 39 | Dauerdüngungsversuch IOSDV | Dülmen | 12 | 3 | medium | Fertilization |
| 40 | Zuckerrübenfruchtfolgeversuch | Etzdorf | 13 | 1 | very high | Fertilization/Crop rotation/Other |
| 41 | Dauerdüngungsversuch (Zuckerrübenmonokultur) | Etzdorf | 13 | 1 | very high | Fertilization/Crop rotation |
| 42 | Dauerdüngungsversuch Getreide (Getreidedauerversucht) | Etzdorf | 13 | 1 | very high | Fertilization/Crop rotation |
| 43 | Dauerdüngungsversuch Getreide (Getreidedauerversuch zur Bekämpfung der Halmbruchkrankheit) | Etzdorf | 13 | 1 | very high | Fertilization/Crop rotation |
| 44 | N-Formen-Versuch | Freising | 14 | 4 | high | Fertilization/Crop rotation |

| 45 | P-Düngung | Freising | 14 | 4 | high | Fertilization |
|----|-----------|----------|----|----|------|---------------|
| 47 | Stroh/Stalldung-Fruchtfolge | Freising | 14 | 4 | high | Fertilization |
| 48 | N-Düngung/Fruchtfolge | Freising | 14 | 4 | high | Fertilization |
| 49 | N-Steigerung mit Kalkstickstoff | Freising | 14 | 4 | high | Fertilization |
| 50 | Versuch 020 N-Formen-Versuch | Freising | 14 | 3 | high | Fertilization |
| 51 | Bodenbearbeitungsversuch Südzucker | Friemar | 15 | 2 | very high | Tillage |
| 52 | Erschöpfungsversuch (EV) | Gießen | 16 | 2 | low | Fertilization |
| 53 | Kalkdüngungsversuch | Gießen | 16 | 2 | high | Fertilization |
| 54 | Dauerversuch Biologische Stickstofffixierung (BSG) | Gießen | 16 | 2 | high | Fertilization/Crop rotation |
| 55 | Ökologischer Ackerbauversuch Gladbacherhof | Villmar | 17 | 2 | extremely low | Fertilization/Tillage/Crop rotation |
| 56 | Bodenbearbeitungsversuch Hohes Feld | Nörten-Hardenberg | 18 | 3 | high | Tillage |
| 57 | Garte-Süd-Bodenbearbeitung (Reinshof) | Göttingen | 18 | 2 | very high | Tillage |
| 58 | Garte-Nord-Bodenbearbeitung (Reinshof) | Göttingen | 18 | 2 | high | Crop rotation |
| 59 | Langzeitversuch zur P- und K-Düngung auf dem Reinshof | Nörten-Hardenberg | 19 | 2 | high | Fertilization |
| 60 | Bodenbearbeitungsversuch Südzucker | Grombach | 15 | 3 | high | Tillage |
| 61 | Kastenparzellenversuch Sandboden / Lehmboden / Tonboden | Großbeeren | 20 | 1 | | Fertilization |
| 62 | PK-Mangelversuch | Groß Gerau | 16 | 1 | very low | Fertilization |
| 63 | Dauerfeldversuch P60 | Groß Kreutz | 21 | 1 | low | Fertilization |

| 64 | Dauerfeldversuch M4 | Groß Kreutz | 21 | 1 | very low | Fertilization |
| 65 | Versuchsfeld der Versuchsstation Groß Lüsewitz | Groß Lüsewitz | 22 | 2 | very low | Other |
| 66 | Ewiger Roggen | Halle | 23 | 1 | medium | Fertilization/Crop rotation |
| 67 | Schmalfuss'scher Dauerversuch, Feld A, Kalkdüngung | Halle | 23 | 1 | very high | Fertilization |
| 68 | Schmalfuss'scher Dauerversuch, Feld C, Kaliumdüngung | Halle | 23 | 1 | very high | Fertilization |
| 69 | Schmalfuss'scher Dauerversuch, Feld D, Phosphordüngung | Halle | 23 | 1 | very high | Fertilization |
| 70 | Organische Düngung (Feld F) | Halle | 23 | 1 | very high | Fertilization |
| 71 | Dauerfeldversuch "Bodenfruchtbarkeit" | Hennef | 7 | 3 | very high | Fertilization |
| 72 | Dauerversuch Düngung-Fruchtfolge | Renningen | 24 | 4 | medium | Fertilization/Crop rotation |
| 73 | Versuch zur Bodenbearbeitung | Renningen | 24 | 3 | low | Tillage |
| 74 | Dauerdüngungsversuch | Hohenschulen | 25 | 3 | high | Fertilization |
| 75 | Stickstoffversuch "Decline-Versuch" | Hohenschulen | 25 | 3 | medium | Fertilization |
| 76 | Fruchtfolgeversuch | Hohenschulen | 25 | 3 | medium | Fertilization/Crop rotation |
| 77 | N-Düngung zu Wintergerste | Hohenschulen | 25 | 3 | medium | Fertilization |
| 78 | Düngerartenvergleich (Versuch I) | Lauterbach | 23 | 5 | medium | Fertilization/Crop rotation |
| 79 | Kombinationswirkung (Versuch II) | Lauterbach | 23 | 5 | very low | Fertilization |

| 80 | Nährstoffverhältnisversuch | Limburgerhof/Bruch | 26 | 1 | very low | Fertilization |
|----|----------------------------|--------------------|----|---|----------|---------------|
| 81 | Feldwirtschaftsversuch | Limburgerhof/Bruch | 26 | 1 | low | Fertilization |
| 82 | Nährstoffmangelversuch | Limburgerhof | 26 | 1 | low | Fertilization |
| 83 | WW-Fruchtfolgeversuch | Ludwigshafen/Ruchheim | 26 | 1 | low | Fertilization/Crop rotation/Other |
| 84 | Bodenbearbeitungsversuch | Ludwigshafen/Ruchheim | 26 | 1 | high | Fertilization/Tillage |
| 85 | Bodenbearbeitungsversuch | Lüttewitz | 15 | 2 | high | Tillage |
| 86 | Dauerdüngungsversuch L28 | Methau | 27 | 3 | high | Fertilization |
| 87 | Dauerdüngungsversuch (V140) | Müncheberg | 28 | 1 | low | Fertilization |
| 88 | Bodenbearbeitung (V760) | Müncheberg | 28 | 1 | low | Tillage |
| 89 | Modellbetrieb Organischer Landbau, Felder 931 - 934 | Müncheberg | 28 | 1 | low | Other |
| 90 | Kalium-Steigerungsversuch Höckelheim/Südniedersachsen | Northeim/Höckelheim | 29 | 2 | low | Fertilization |
| 91 | P-Düngung auf Sandmischkultur | Oldenburg/Friesoythe | 29 | 3 | medium | Fertilization |
| 92 | Bodenbearbeitung/Fruchtfolge | Oldenburg/Friesoythe | 18 | 3 | extremely low | Tillage/Crop rotation |
| 93 | Bodenbearbeitung | Oldenburg/Friesoythe | 18 | 3 | extremely low | Tillage |
| 94 | Internationaler Organischer Stickstoffdüngungs-Versuch (IOSDV) | Oldenburg | 30 | 3 | extremely low | Fertilization |
| 96 | Dauerversuch 'Auswirkung von Daueranbau' | Puch | 31 | 5 | extremely low | Crop rotation |
| 97 | Verbesserte Dreifelderwirtschaft | Puch | 31 | 5 | high | Crop rotation |
| 98 | Getreide/Mais Fruchtfolge | Puch | 31 | 5 | high | Crop rotation |
| 99 | Einfluss von Grundbodenbearbeitung | Puch | 31 | 5 | high | Tillage |

| 100 | Internationaler Organischer Stickstoffdüngungs-Versuch (IOSDV) | Puch | 31 | 5 | high | Fertilization |
|-----|---------------------------------------------------|-------------------|----|---|--------|---------------------------------|
| 101 | Internationaler Organischer Stickstoffdüngungs-Versuch (IOSDV) | Rauischholzhausen | 16 | 2 | high | Fertilization |
| 102 | Organische Düngung / Stalldung Schafpferchversuch | Rauischholzhausen | 16 | 2 | high | Fertilization |
| 103 | Gründüngung / Strohdüngungsversuch | Rauischholzhausen | 16 | 2 | high | Fertilization |
| 104 | Bilanzversuch Kastenanlage | Rauischholzhausen | 16 | 2 | high | Fertilization |
| 105 | Wirkungen differenzierter Bodenbearbeitungssysteme im Dauerversuch Scheyern | Scheyern | 32 | 4 | high | Fertilization/Tillage/Crop rotation |
| 106 | Fruchtfolgedüngungsversuch | Seehausen | 23 | 1 | high | Fertilization/Crop rotation |
| 107 | Konzentrationsversuch | Seehausen | 23 | 1 | high | Crop rotation |
| 108 | Düngungs-Kombinationsversuch Seehausen (F1-70) | Seehausen | 23 | 1 | high | Fertilization |
| 109 | Bodenbearbeitungsversuch | Seehausen | 23 | 1 | high | Tillage |
| 110 | Gülledauerversuch | Seehausen | 23 | 1 | high | Fertilization |
| 111 | Bodenfruchtbarkeitsversuch | Seehausen | 23 | 1 | high | Fertilization/Tillage |
| 112 | Internationaler Organischer Stickstoffdüngungs-Versuch (IOSDV) | Speyer | 33 | 2 | high | Fertilization/Tillage |
| 113 | Humusversuch | Speyer | 33 | 2 | medium | Fertilization/Other |
| 114 | Kali-Magnesium-Kalk-Versuch | Speyer | 33 | 2 | medium | Fertilization |
| 115 | Klärschlammversuch | Speyer | 33 | 2 | medium | Other |

| 116 | Bracheversuch | Speyer | 33 | 2 | medium | Other |
|-----|---------------|--------|-----|-----|--------|-------|
| 117 | Dauerdüngungsversuch L28 | Spröda | 27 | 1 | medium | Fertilization |
| 119 | Düngungs- und Beregnungsversuch (Thy_D1) | Thyrow | 34 | 1 | high | Fertilization/Other |
| 120 | Stroh- und N-Düngung in Fruchtfolgen mit unterschiedlichem Getreideanteil (Thy_D5) | Thyrow | 34 | 1 | very low | Fertilization/Crop rotation |
| 121 | Statischer Nährstoffmangelversuch (Thy_D41) | Thyrow | 34 | 1 | very low | Fertilization |
| 122 | Nährstoffmangelversuch Winterroggen Monokultur (Thy_D42) | Thyrow | 34 | 1 | very low | Fertilization |
| 123 | Statischer Bodenfruchtbarkeitsversuch (Thy_D6) | Thyrow | 34 | 1 | very low | Fertilization |
| 125 | Strohdüngungsversuch (Thy_D2) | Thyrow | 34 | 1 | very low | Fertilization |
| 136 | Modellbetrieb Organischer Landbau, Felder 901 - 904 | Müncheberg | 28 | 1 | very low | Other |
| 137 | Statischer Dauerfeldversuch "organisch-mineralische N-Düngung" | Großbeeren | 20 | 1 | | Fertilization |
| 138 | Versuch zur Bodenbearbeitung | Schönberg | 35 | 3 | low | Tillage |
| 139 | Gehölzhäckselapplikation | Schönberg | 35 | 3 | very low | Other |
| 140 | Versuch 700 (Reduzierte Bodenbearbeitung) | Schönberg | 35 | 3 | extremely low | Tillage |
| 142 | Effiziente Nährstoffverwertung, K- | Pommritz | 27 | 2 | extremely low | Fertilization |

| | | | | | | |
|---|---|---|---|---|---|---|
| | Eichversuche | | | | | |
| 143 | Effiziente Nährstoffverwertung, K-Eichversuche | Forchheim | 27 | 4 | extremely low | Fertilization |
| 144 | Referenzfläche | Hennef | 7 | 3 | medium | Fertilization |
| 146 | Statischer Versuch Bodennutzung (Thy_D3/1) | Thyrow | 34 | 1 | very low | Fertilization/Tillage |
| 147 | Statischer Dauerfeldversuch Organische Düngung und Humusreproduktion (Thy_D3/2) | Thyrow | 34 | 1 | medium | Fertilization |
| 148 | Statischer N-Düngungsversuch in Winterroggen-Monokultur (Thy_D7) | Thyrow | 34 | 1 | very low | Fertilization |
| 149 | Alte dreifeldrige Fruchtfolge | Puch | 31 | 5 | very low | Fertilization/Crop rotation |
| 150 | Fruchtfolgen im ökologischen Landbau | Puch | 31 | 5 | very low | Fertilization/Crop rotation |
| 151 | Fruchtfolgen im ökologischen Landbau | Viehhausen | 31 | 4 | high | Fertilization/Crop rotation |
| 152 | Fruchtfolgeversuch (FF) | Rauischholzhausen | 16 | 2 | high | Crop rotation |
| 153 | Bodenbearbeitungs-Versuch (BB) | Rauischholzhausen | 16 | 3 | high | Tillage |
| 154 | Bodenbearbeitungsversuch Südzucker | Zschortau | 15 | 1 | high | Tillage |
| 155 | Bodenbearbeitungsversuch Südzucker | Insultheim | 15 | 2 | high | Tillage |
| 156 | Bodenbearbeitungsversuch Südzucker | Sailtheim | 15 | 3 | high | Tillage |
| 157 | Bodenbearbeitungsversuch | Gieshügel | 15 | 2 | medium | Tillage |

Südzucker

| 158 | Strategievergleich umweltschonender Pflanzenschutz (BS1) | Dahnsdorf | 10 | 1 | low | Other |
|-----|-----|-----|-----|-----|-----|-----|
| 159 | Ökologischer Landbau (öko1) | Dahnsdorf | 10 | 1 | high | Other |
| 160 | Strategien zur Minderung der Anwendung chemischer Pflanzenschutzmittel (BS4) | Dahnsdorf | 10 | 1 | high | Other |
| 161 | Kalk-Düngungsversuch | Weilmünster-Ernsthausen | 36 | 3 | high | Fertilization |
| 162 | Phosphordüngungsstrategien | Biestow | 37 | 2 | high | Fertilization |
| 165 | Körnermais Daueranbau | Rotthalmünster | 38 | 3 | extremely low | Fertilization |
| 166 | Winterweizen Daueranbau | Rotthalmünster | 38 | 3 | medium | Other |
| 167 | E-Feld (bis 1957) | Göttingen | 18 | 3 | medium | Fertilization |
| 193 | Dauerfeldversuch (DE-1b-F-1, Am Kotten) | Rosendahl Holtwick | 12 | 3 | medium | Fertilization |
| 194 | Dauerfeldversuch (DE-1b-F-2, Am Hof) | Dülmen Karthaus | 12 | 3 | no data | Fertilization |
| 195 | Dauerfeldversuch (DE-1b-F-3, IPU Schlag 9) | Dülmen | 12 | 3 | medium | Fertilization |
| 197 | Feldmodellversuch "Krumenaufbau" | Müncheberg | 28 | 1 | medium | Fertilization/Tillage |
| 203 | Kalkformenversuch | Cunnersdorf | 39 | 3 | medium | Fertilization |
| 205 | Dauerdüngungsversuch (M70) | Groß Kreuz | 40 | 1 | low | Fertilization |
| 206 | Getreidedauerversuch | Noitzsch | 13 | 1 | very low | Fertilization/Crop rotation/Other |
| 207 | Stroh-Stallmistversuch | Lentföhrden | 25 | 3 | very low | Fertilization |
| 208 | Phosphor-Steigerungsversuch | Schädtbek | 25 | 2 | very low | Fertilization |
| 209 | Fruchtfolgeversuch | Gülzow | 41 | 2 | medium | Fertilization/Tillage |

| | | | | | | |
|---|---|---|---|---|---|---|
| | Bodenbearbeitung/organische Düngung Winterraps (FF 1.1) | | | | | |
| 210 | Fruchtfolgeversuch Bodenbearbeitung/organische Düngung Sommerweizen (FF 1.2) | Gülzow | 41 | 2 | medium | Fertilization/Tillage |
| 211 | Fruchtfolgeversuch Bodenbearbeitung/organische Düngung Winterweizen (FF 2.1) | Gülzow | 41 | 2 | medium | Fertilization/Tillage |
| 212 | Fruchtfolgeversuch Bodenbearbeitung/organische Düngung Silomais (FF 2.2) | Gülzow | 41 | 2 | medium | Fertilization/Tillage |
| 213 | Schmalfuss'scher Dauerversuch, Feld B (physiologischen Reaktion von Düngemitteln) | Halle | 23 | 1 | medium | Fertilization |
| 214 | Schmalfuss'scher Dauerversuch, Feld E, Stickstoffdüngung | Halle | 23 | 1 | medium | Fertilization |
| 217 | E-Feld (ab 1957) | Göttingen | 18 | 3 | very high | Fertilization |
| 218 | Modellversuch zur Bodenbildung | Halle | 23 | 1 | very high | Fertilization |
| 219 | Weihenstephaner Kali-Formenversuch | Weihenstephan | 30 | 4 | no data | Fertilization |
| 220 | Kleinparzellenversuch Hu1 bzw. Hu1To9 | Rostock | 37 | 2 | no data | Fertilization |
| 221 | Organische Düngestoffe - Wirkung  (V140/06) | Dedelow | 28 | 1 | low | Fertilization |
| 222 | Organische Düngestoffe - | Dedelow | 28 | 1 | low | Fertilization |

| | | | | | | |
|---|---|---|---|---|---|---|
| 223 | Organische Düngestoffe - Wirkung (V140/08) | Dedelow | 28 | 1 | low | Fertilization |
| 224 | Organische Düngestoffe - Wirkung (V140/09) | Dedelow | 28 | 1 | low | Fertilization |
| 225 | Bodenbearbeitungsversuch am Galgenberg | Bingen-Büdesheim | 42 | 1 | very low | Tillage/Other |
| **Grassland LTFE** | | | | | | |
| 10 | Stickstoffdüngung auf Grünland | Iden | 6 | 1 | | Fertilization |
| 12 | Stickstoffdüngung auf Grünland | Hayn | 6 | 3 | | Fertilization |
| 32 | Schachbrettversuch / Dauerdüngungsversuch auf Grünland | Daun | 7 | 4 | | Fertilization |
| 46 | K-, P-, N-Steigerung zu Grünland | Freising | 14 | 4 | | Fertilization |
| 95 | Grünlanddauerversuch (V102) | Paulinenaue | 28 | 1 | | Fertilization |
| 118 | P-Düngungsversuch | St. Peter | 36 | 5 | | Fertilization |
| 135 | Grünlandversuch Weiherwiese | Steinach | 31 | 3 | | Fertilization |
| 141 | Kalk-Düngungsversuch | Rösrath | 36 | 4 | | Fertilization |
| 163 | Grünlandversuch Veitshof | Veitshof | 43 | 3 | | Fertilization |
| 164 | Statischer Dauerdüngungsversuch | Rotthalmünster | 38 | 3 | | Fertilization |
| 168 | Phosphordüngung auf Grünland | Christgrün | 27 | 3 | | Fertilization |
| 169 | Kaliumdüngung auf Grünland | Christgrün | 27 | 3 | | Fertilization |
| 170 | Phosphordüngung auf Grünland | Forchheim | 27 | 4 | | Fertilization |
| 171 | Kaliumdüngung auf Grünland | Forchheim | 27 | 4 | | Fertilization |

| 172 | Phosphordüngung auf Grünland | Hayn | 6 | 3 | Fertilization |
|-----|------------------------------|------|---|---|---------------|
| 173 | Kaliumdüngung auf Grünland | Hayn | 6 | 3 | Fertilization |
| 174 | Phosphordüngung auf Grünland | Iden | 6 | 1 | Fertilization |
| 175 | Kaliumdüngung auf Grünland | Iden | 6 | 1 | Fertilization |
| 176 | Phosphordüngung auf Grünland | Oberweißbach | 44 | 5 | Fertilization |
| 177 | Kaliumdüngung auf Grünland | Oberweißbach | 44 | 5 | Fertilization |
| 178 | Überprüfung der Kalkempfehlung für Grünland | Christgrün | 27 | 3 | Fertilization |
| 179 | Umweltbewusste Grünlandbewirtschaftung | Christgrün | 27 | 3 | Fertilization/Other |
| 180 | Grunddüngung im Grünland | Christgrün | 27 | 3 | Fertilization |
| 181 | Phosphordüngung auf Grünland | Heßberg | 44 | 3 | Fertilization |
| 182 | Kaliumdüngung auf Grünland | Heßberg | 44 | 3 | Fertilization |
| 183 | Phosphordüngung auf Grünland | Paulinenaue | 21 | 1 | Fertilization |
| 184 | Kaliumdüngung auf Grünland | Paulinenaue | 21 | 1 | Fertilization |
| 185 | Phosphordüngung auf Grünland | Wechmar | 44 | 2 | Fertilization |
| 186 | Kaliumdüngung auf Grünland | Wechmar | 44 | 2 | Fertilization |
| 187 | Niederblockland | Bremen | 45 | 2 | Fertilization |
| 188 | Kalkbedarf der Hochmoorkulturen | Bremen | 45 | 3 | Fertilization |
| 189 | Königsmoor/Nordheide | Bremen | 45 | 3 | Fertilization |
| 198 | Versuch 250 (Nährstoffmangelversuch) | Ihinger Hof | 46 | 4 | Fertilization |
| 199 | Versuch 251 | Ihinger Hof | 46 | 4 | Fertilization |

Institutional addresses:

Landwirtschaftskammer Rheinland-Pfalz, Referat 21 Pflanzenbau, Burgenlandstr. 7, 55543 Bad Kreuznach

Helmholtz-Zentrum für Umweltforschung, Versuchsstation Bad Lauchstädt, Hallesche Straße 44, 06246 Bad Lauchstädt

Thüringer Landesamt für Landwirtschaft und Ländlichen Raum, Postfach 100 262, 07702 Jena

Humboldt-Universität zu Berlin, Lehr- und Forschungsstation, Bereich Pflanzenbauwissenschaften, Albrecht-Thaer-Weg 5, 14195 Berlin

Hochschule Anhalt, Fachbereich Landwirtschaft, Strenzfelder Allee 28, 06406 Bernburg

Landesanstalt für Landwirtschaft und Gartenbau, Strenzfelder Allee 22, 06406 Bernburg

Universität Bonn, Institut für Nutzpflanzenwissenschaften und Ressourcenschutz, Karlrobert-Kreiten-Str. 13, 53115 Bonn

Julius Kühn-Institut, Bundesforschungsinstitut für Kulturpflanzen, Messeweg 11/12, 38104 Braunschweig

Stabsstelle Boden des Thünen-Instituts, Bundesallee 50, 38116 Braunschweig

Julius Kühn-Institut, Bundesforschungsinstitut für Kulturpflanzen, Stahnsdorfer Damm 81, 14532 Kleinmachnow

Forschungsring e.V., Brandschneise 5, 64295 Darmstadt

YARA GmbH & Co. KG, Hanninghof 35, 48249 Dülmen

Martin-Luther-Universität Halle-Wittenberg, Allgemeiner Pflanzenbau / Ökologischer Landbau, Betty-Heimann-Str. 5, 06120 Halle

Technische Universität München, Lehrstuhl für Pflanzenernährung, Emil-Ramann-Straße 2, 85354 Freising

Institut für Zuckerrübenforschung an der Georg-August-Universität Göttingen, Holtenser Landstr. 77, 37079 Göttingen

Justus-Liebig-Universität Gießen, Institut für Pflanzenbau und Pflanzenzüchtung I, Biomedizinisches Forschungszentrum Seltersberg, Schubertstraße 81, 35392 Gießen

Justus-Liebig-Universität Gießen, Lehr- und Versuchsbetrieb für ökologischen Landbau, 65606 Villmar

Georg-August-Universität Göttingen, Abteilung Pflanzenbau, Von-Siebold-Str.8, 37075 Göttingen

Georg-August-Universität Göttingen, Fakultät für Agrarwissenschaften, Carl-Sprengel-Weg 1, 37075 Göttingen

Leibniz-Institut für Gemüse- und Zierpflanzenbau, Theodor-Echtermeyer-Weg 1, 14979 Großbeeren

Landesamt für Verbraucherschutz, Landwirtschaft und Flurneuordnung, Referat Ackerbau und Grünland, Berliner Straße, 14532 Stahnsdorf, OT Güterfelde

Julius Kühn-Institut, Bundesforschungsinstitut für Kulturpflanzen, Rudolf-Schick-Platz 3a, OT Groß Lüsewitz, 18190 Sanitz

Martin-Luther-Universität Halle-Wittenberg, Naturwissenschaftliche Fakultät III, Institut für Agrar- und Ernährungswissenschaften, Lehr- und Versuchsanstalt Halle, 06099 Halle (Saale)

Universität Hohenheim, Versuchsstation Agrarwissenschaften, Standort Ihinger Hof (403), 71272 Renningen

Christian-Albrechts-Universität zu Kiel, Abteilung Acker- und Pflanzenbau, Institut für Pflanzenbau und Pflanzenzüchtung, Hermann-Rodewald-Str. 9, 24118 Kiel

Agrarzentrum Limburgerhof, Speyerer Str. 2, 67117 Limburgerhof

Sächsisches Staatsministerium für Energie, Klimaschutz, Umwelt und Landwirtschaft, Abteilung 7, Referat 72, Wilhelm-Buck-Straße 2, 01097 Dresden

Leibniz-Zentrum für Agrarlandschaftsforschung, Eberswalder Str. 84, 15374 Müncheberg

Landwirtschaftskammer Niedersachsen, Bezirksstelle Hannover, Wunstorfer Landstr. 11, 30453 Hannover

unbekannt

Landesanstalt für Landwirtschaft, Institut für Ökologischen Landbau, Bodenkultur und Ressourcenschutz, Lange Point 6, 85351 Freising

Versuchsstation Klostergut Scheyern, 85298 Scheyern

Landwirtschaftliche Untersuchungs- und Forschungsanstalt Speyer, Obere Langgasse 40, 67346 Speyer

Humboldt-Universität zu Berlin, Lehr- und Forschungsstation Pflanzenbauwissenschaften, Thyrower Dorfstraße 9, 14959 Trebbin, OT Thyrow

Universität Hohenheim, Kleinhohenheim 1, 70599 Stuttgart-Schönberg

FEhS-Institut für Baustoff-Forschung e.V. (Forschungsgemeinschaft Eisenhüttenschlacken), Bliersheimer Straße 62, 47229 Duisburg

Universität Rostock, Agrar- und Umweltwissenschaftliche Fakultät, Justus-von_Liebig-Weg 6, 18059 Rostock

Staatliche Höhere Landbauschule Rotthalmünster, Franz-Gerauer-Straße 22-24, 94094 Rotthalmünster

SKW Stickstoffwerke Piesteritz GmbH, Möllensdorfer Straße 13, 06886 Lutherstadt Wittenberg

Landesamt für Ländliche Entwicklung, Landwirtschaft und Flurneuordnung, Neue Chaussee 3A, 14550 Groß Kreutz (Havel)

Landesforschungsanstalt für Landwirtschaft und Fischerei Mecklenburg-Vorpommern, Dorfplatz 1 / OT Gülzow, 18276 Gülzow-Prüzen

Technische Hochschule Bingen, Team Landwirtschaft und Umwelt, Berlinstraße 109, 55411 Bingen am Rhein

Technische Universität München, Lehrstuhl für Grünlandlehre, Alte Akademie 12, 85354 Freising

Thüringer Landesamt für Landwirtschaft und Ländlichen Raum, Naumburger Str. 98, 07743 Jena

Landesamt für Bergbau, Energie und Geologie, Stilleweg 2, 30655 Hannover

Universität Hohenheim, Fachgebiet Nachwachsende Rohstoffe in der Bioökonomie (340b), Fruwirthstraße 23, 70599 Stuttgart

630