# Peer review of "Long-term Field Experiments in Germany: Classification and spatial Representation"

_SOIL, 2020_

## Short Comment (SC1) · 19 Jun 2020

Mueller 2010 not in list. BGR 2007 and 2014 in list but with a long URL as "author".

---

## Short Comment (SC2) · 19 Jun 2020

Thank you very much. I just checked out the problems. However, Mueller et al 2010 I did find in the reference list (lines 362-364). For the BGR sources it is true. We will add the BGR as author. Kind regards, Meike Grosse

---

## Short Comment (SC3) · 22 Jun 2020

Dear Meike,

as just discussed by email, it would be nice to have the download links for the open access CWB data added to the paper.

Kind regards,

Ute

---

## Short Comment (SC4) · 23 Jun 2020

Dear Ute,

Thank you for the comment. I will add the download link for the CWB data to the paper.

Best regards,

Meike
* * *

---

## Referee Comment (RC1) · Anonymous Referee #1 · 2 Jul 2020

This paper presents an interesting analysis on long-term field experiments (LTFEs) and their representativeness in Germany. The paper shows the diversity of the experiments and that they cover most climatic water balance classes and also soil quality classes. This is an interesting result. The paper is written well and is nicely structured. For the international readership of SOIL it might be of limited interest, since all results are related to Germany without direct implications for outside Germany.

The most striking challenge is the data availability. So far, the work presented in this study on a common database for German LTFEs was mainly concerned with compiling meta data from the LTFEs. This is of great value that will be appreciated by the scientific community. However, there is no clear plan how to make all or most data from the LTFEs accessible and open access also to the international soil and agricultural

science community. Out of more than 200 LTFEs only two LTFEs were indicated in the map as "data available" and only seven additional sites with "data available soon". In the paper the great value of combined analysis of many LTFEs is described (e.g. l. 23 f). However, if the data of the LTFEs, such as soil properties and yield, are not available, not harmonized, not quality checked and not open accessible, such a combined analysis is not possible. The work presented in this study is a first step towards such analysis. However, it needs to be outlined how the next steps towards full data accessibility will be achieved for how many LTFEs. Without the next step the first step is of limited value. In l. 18 it is claimed that the presented database are an important step to provide access to the research data. More details need to be outlined how this access will be provided in the future. The one sentence in l. 68 ("There is a focus on reseach data from LTFEs") is not enough.

l. 136: This study shall be published in a soil science journal. However, in particular the representativeness analysis for soils in rather incomplete. It is restricted to one soil quality indicator and this indicator was available only for croplands. Thus, it is missing for 28% of the LTFEs. Maybe more abundant soil data, such as texture or soil type, can be used for classification and the representativeness analysis. Texture is the major soil parameter that can hardly be changed with management and influences all soil processes and plant growth. In l. 267 a study is mentioned that found under-representation of LTFEs on clayey sites. This result cannot be compared with the recent study since such an analysis is missing and would significantly increase the value of this study.

Additional comments: l. 6: Soil monitoring of climate impact can be performed much more cost efficient on permanent sampling sites (such as "Bodendauerbeobachtung"). Since LTFEs do not represent real practice field sites they might miss some trends that can only be monitored at farmers' field sites. The value of LTFEs is to provide data on management impacts (under changing climate).

L 16: The representation and distribution of management options in the LTFEs is missing as a result in the abstract. Since this is the main aim of LTFEs it would be worth to include one or two sentences on how management treatments are covered in LTFEs in Germany.

l. 28: In agriculture, plant nutrition is linked to fertilisation. Thus, these are not two but one and the same aspect.

l. 39: The definition of "control treatments" is not clear. Is the control treatment defined by each LTFE or does it depend on the study? Customary or common management practices are changing over time e.g. the fraction of reduced tillage or fertilisation type and amount. Is the control treatment than also changing over time?

l. 45: Change "landscapes to "soil" since LTFEs does not comprise landscapes.

l. 99 and 102: Why 191? 94+87=181

l. 156: It is not comprehensible why many grassland LTFEs were excluded. This need to be explained and justified since grassland trials are under-represented in the compiled LTFE dataset. Above it is written that LTFEs are useful beyond the original scope or research theme. Here it is argued that the research theme of the grassland trials did not fit and were therefore excluded.

l. 192: What is a technical college? A university of applied sciences?

l. 200-206: This section is redundant and repetition from above an can be removed.

l. 214-l. 223: For an international readership of the journal, it would be good to provide a map with the names of the regions mentioned here or include the names in Fig 5.

Fig. 3: The colours are not easy to distinguish, in particular that for tillage, fertilisation and crop rotation.

Fig 5 and 6: The dispersion of points from only single experimental sites with different experiments results in biased impressions, e.g. that the whole region of Halle is covered with LTFEs even though there might be only one single experimental site. I

propose to either strongly reduce the dispersal of the points from one site or completely avoid them since this map aims at illustrating the spatial distribution and representativeness of LTFEs and one site with many trails mostly does not contribute in achieve a higher representativeness of soils and climate.

Fig 5: The map seems to be incomplete for German agricultural land (with is the reference for this study). Mostly grassland seem to be missing, e.g. in the pre Alps, the Sauerland or in North-Western Germany. Readers expect that the class "other land" comprise only non-agricultural land. Maybe CORINE data are not appropriate but ATKIS Basis DLM data can be used.

Fig. 7: This illustration with boxes is unusual and thus difficult to read. Since the y-axis contains distinct values (no classes) a representation with points or lines would be more appropriate.

---

## Referee Comment (RC2) · Anonymous Referee #2 · 3 Jul 2020

Grosse et al. compiled an overview over historic and still running long-term experiments in Germany. I applaud this effort because it will allow better use of these expensive and precious experiments.

However, their assessment could be more critical and identify major deficits of the present LTFEs. As far as I can see, all are situated in flat areas (a data evaluation in this respect would be nice and not too difficult to do). This means that they exclude major lateral processes (interflow, surface runoff) and differ largely from typical agricultural fields, in contrast to the conclusions from the climatic water balance during the growing season (CWBg) and the Müncheberger Soil Quality Rating (MSQR) that seem to indicate representativeness. This deficit may be especially pronounced for grassland experiments because grassland either occupies lowland areas that are too wet

for arable use or areas that are too steep. For grassland experiment, which in fact are meadow experiments (grazings seems to be missing; also a major deficit). Such critical assessment would be extremely helpful to guide the installation of future LTFEs and to show the limitations in the conclusions that can be drawn from the existing LTFEs.

Were lysimeter experiments included, which would allow assessing at least vertical water fluxes? Do long-term experiments with lysimeter exist at all in Germany? Again, a critical assessment would be helpful. Were experiments included that allow quantification of lateral processes (runoff, soil loss)? I could imagine that the measurements in Trier (Stehling and Schmidt 2017) or those by Jung and Brechtel (1980) qualify for LTFE. If they don't qualify, this would again illustrate a major deficit of present LTFEs.

In the discussion I missed a wider view. Do similar compilations also exist in other countries? Are the German LTFE experiments similar to what was done and is done in other countries?

Furthermore, the authors give the impression that they still focus on the old questions of LTFEs (mainly yield) that became boring. I had this impression for two reasons. First, little examples are given how LTFEs can be used in fascinating modern research on urgent questions. Second, using LTFEs in modern research applying new techniques requires access to the experiments. Hence it makes a big difference whether an experiment is still ongoing or not. However, this information is given nowhere. Second, it often requires archived samples (as an example what can be done with modern techniques and archived samples, Köhler et al. 2012 comes to my mind but there are certainly more examples). This information, whether archived samples are available, should be included. Generally, I missed information about which data could be obtained from the LTFEs.

Most of my other remarks are mainly editorial issues. The weakest part in this respect is the table in the Appendix, which is most important because it resolves the LTFEs and thus allows access (see below).

Details (numbers refer to lines):

12: add "during the growing season"; I would even change the abbreviation to CWBg because usually an entire year is considered in a CWB. I was very surprised when suddenly somewhere in the manuscript the information 'growing season' popped up

13: Müncheberger Soil Quality Rating seems to be a combination of German and English. Shouldn't it be 'Müncheberg'?

35: I welcome this definition of the control that is certainly better than the often used but wrong assignment of the strongest and most unrealistic intervention as control, namely the long-term nutrient removal. However, I did not find this definition to be used later in the manuscript.

46: Bai et al.

116: Not clear how PET was derived. Was it taken from DWD? Is it Haude?

126: This is strange. Later only 6 classes of the MSQR are used, not 102. I wonder whether different properties like soil structure, wetness, relief, contaminations can be combined in one indicator of six classes. This may be possible for one specific target like yield but will fail for most other targets or require other classes. Is a better resolution than these six classes possible?

128: I guess this should read 'available water capacity'

130: What is unsuitable? This always requires the definition of a target.

139: This leads to the question: Were lysimeter experiments included? If not, why not?

155: The title does not have this restriction; also the Abstract does not. I wonder why it suddenly pops up in the results. I also wonder how this is defined (what is bioeconomy?) and whether these experiments really aim at sustainable soil use. They exclude many things that make soil use unsustainable (erosion, compaction) and hence are unsuitable to test sustainability (in this general sense). I also wonder even more why

the criterion sustainability excludes some grassland experiments. This is contrary to what I would expect.

160: Establishment was in the past. Hence past tense would be appropriate. The question of correct tense is rather difficult to answer given that 30% of the experiments have come to an end already and others will come to an end in the future, I wonder whether the mostly used present tense is justified.

171-172: One sentence is usually not a paragraph. Furthermore, temporal aspects were treated in the first paragraph of the results. I suggest moving this sentence.

173: sentences usually do not start with a number; this also applies in other cases (e.g. L. 181, 184).

178 : Move opening parenthesis

208-209: This should be moved to the M & M section; this is the first time that growing period is mentioned although CWB appeared already several times. Furthermore, it would be good to explain the rationale behind this decision than let the reader speculate

266-269: I would reverse the argument. In my view the critique by Franko is well justified and shows that 6 classes of the MSQR are insufficient. I do not suggest to include an assessment of the complexity of soil parameters but it is also not justified do say that the LTFEs are representative regarding soils just because they match the rather coarse and restricted (to yield) MSQR criterion. References: The format varies among references. Please homogenize

Fig. 2: The pie charts are an attempt to illustrate the manuscript. However, they do a poor job. They require a legend, which is difficult to read (because font size is smaller than that of ordinary text) and contain information that is better suited for a table or even could be given as plain test. For Fig. 2 a, a density graph would be more appropriate

Fig. 3: A graph usually has not a title but a caption. The colors are impossible to distinguish Are they necessary? Can they be simplified? Wouldn't the year when an

LTFE was closed be equally interesting?

Table 1: It is not clear whether 'organic fertilization' also includes straw and compost (there is not an equivalent 'Mineral fertilization'). Furthermore, why are green manure, compost and sludge mentioned, but not the main type of organic manure. This classification appears inconsistent. It surprises me that only two of the grassland experiments have organic fertilizer although grassland use unavoidably produces manure. Have all except for two experiments used an unrealistic design that does not allow application of the results to typical situations? Better call 'plant protection' 'crop protection'

Fig. 4: same remark as Fig. 2

Table 2 + 3: 'vegetation period' should not be in the column head but in the caption. Also the lines separating groups of variables are not consistent (why are CWB class and range separated by a line? Isn't the unit for CWB mm/yr?

Fig. 5: Here four classes of LTFE are sufficient. Why does Fig. 3 require eight classes (that cannot be read anyhow)? LTFE should not be repeated five times in the legend. It is not necessary at all. CWB is in mm/yr

Fig. 6: Delete LTFE

Fig. 7: column widths could be much smaller while larger row heights would allow a larger font size. Presently the numbers hardly can be read. It is not necessary repeating 'MSQR class' six times. Better use a larger font size. The colors of the legend should agree with the colors in the graph.

Table A 1: This is likely the most important table because it allows access to the LTFEs. However, it is rather inconsistent and difficult to read. E.g., the IDs cannot be read; some institutions got abbreviations (why?) others not; some places are mentioned, others not (why?). Mentioning the main institution may be fine in hierarchical organizations but this is clearly insufficient for big universities. Whom should one ask there? I suggest replacing the information in column 3 by a number and the place and

resolving the number below the table by reporting the full addresses. This would also create room for the other columns. Furthermore, I see no reason why umlauts are replaced. This is poor technology of the past century and again a waste of space.

Jung, L. & Brechtel, R. (1980): Messungen von Oberflächenabfluß und Bodenabtrag auf verschiedenen Böden der Bundesrepublik Deutschland. – Schriftenreihe des Deutschen Verbandes für Wasserwirtschaft und Kulturbau (DVWK), 48; Parey, Hamburg

Köhler I, Macdonald A, Schnyder H (2012) Nutrient supply enhanced the increase in intrinsic water-use efficiency of a temperate seminatural grassland in the last century. Global Change Biology 18, 3367–3376, doi: 10.1111/j.1365-2486.2012.02781.x

Stehling, E. & Schmidt, R.-G. (2000/2017): Das Datenarchiv der Forschungsstelle Bodenerosion in Mertesdorf (Ruwertal). Eine Dokumentation über 25 Messjahre (1974 - 1999); Informationszusammenstellung zum Gebrauch der Daten-CD.

---

## Referee Comment (RC3) · Anonymous Referee #3 · 17 Jul 2020

I am very pleased that the German long-term field experiments (LTFE) are now being presented to an international readership in this comprehensive manner. They have a long history and many of the findings made there, as well as the long-term data series, are of great value to the international soil science community and also allow a better understanding of the temporal dynamics of agricultural land. I therefore welcome this manuscript and believe that it is excellently placed in the journal SOIL. Since the journal has open peer review, I can dispense with the repetition of aspects already mentioned in the previous reviews. Overall, I support all comments made there, with the exception that, unlike reviewer #1, I consider this manuscript to be highly interesting for the international readership.

General comments:

[Figure]

The Material and Methods chapter explains how the geospatial analysis is done and also the classification criteria for the LTFEs. However, there is no information on how the experimental design should be analyzed as stated as one of the two main objectives of this study. Do statistical methods come to use? Which ones? The pure assignment of LTFPs to four different classes (five in table 1 and eight in figure 3?) without further statistical analyses (e.g. various types of discriminant analysis, contingency and cross tabulation, factor analysis) is not very appealing. The same holds true for the analysis of the data for climate (CWB) and soil fertility (MSQR) given as number of cases and percentage of share of classes (tables 2 and 3). I am convinced that the manuscript would greatly benefit from a profound statistical analysis and that this would allow (i) a critical discussion of the value of the data that exist so far and (ii) to conclude how such laborious and expensive experiments could be designed in future. A purely qualitative, merely descriptive analysis has certainly been carried out to a sufficient extent in the large number of papers already published on this subject, most of them mentioned generously. A discussion of the results including international literature and experiences of long-term experiments, e.g. from England, China or the US, is missing to a large extent. I recommend that the discussion be significantly revised and expanded in these points. Appropriate quantitative methods for the analysis of the experimental design and the spatial distribution of the experiments with regard to climate and soil fertility should be added. Summarizing, the manuscript appears to be immature in itself and its analytical stringency needs to be improved. Since I consider the topic timely and of high scientific importance, I would be pleased if the authors would thoroughly revise the manuscript.

Specific comments:

Line 49-55: the enumeration of the number of LTFEs published over the years by Cherries seems unnecessary in this way. If the details here are important I would recommend to present it as a table.

Lines 63-80: after the objectives of the work have been formulated in lines 61-63, the

explanations given here seem like a description of material and methods. I recommend to shorten this part and to integrate it into the chapter Material and Methods.

Line 68: what is meant by research parameters? Please list.

Line 83: after the explanations in the introduction regarding the work on the German LTFEs prepared by Koerschens et al., it seems incomprehensible why a new literature study should be made here and would require a corresponding justification. This should also explain why the work of Koerschens et al. is obviously not adequate to follow the objectives of this study.

Line 95: here, too, the technical justification for the selected research topics is missing. Especially with regard to the aspect of a meta-analysis of the research statements, which was prominently emphasized in the introduction, the research topics listed here appear incomplete.

Lines 200-206: the description of the methodology belongs in the corresponding chapter and is superfluous here, as are lines 208 and 209. Similar mixtures of results and material and methods are also shown in the following chapters. I would recommend to check the results part and to concentrate all methodical information at the appropriate place.

Figure 1 does not seem necessary to me, the content is very simple and directly repeats the statements in the text without a gain in information.

The core statements in figure 3 could certainly be presented much more clearly. At the moment most of the space is taken up by the legend. It also seems unusual to me that the figure itself contains a headline ('Start of LTFE').

―――――――――――――――

---

## Author Comment (AC1) · 27 Jul 2020

**Author Response to Anonymous Referee #2**

| Review comment | Author response |
|---|---|
| All LTFE are situated in flat areas (a data evaluation in this respect would be nice and not too difficult to do). This means that they exclude major lateral processes (interflow, surface runoff) and differ largely from typical agricultural fields.
This deficit may be especially pronounced for grassland experiments because grassland either occupies lowland areas that are too wet for arable use or areas that are too steep. | Indeed, lateral processes are typically not analysed in LTFE and they are not designed for such questions. Different design such as the 'Wishmeyer plots' are implemented for erosion studies. We are going to write a section explicitly about deficits in the setup of LTFE and about related experimental setups such as Wishmeyer plots. We also like to write something about the differentiaton from LTFE and soil monitoring sites (BDF). |
| For grassland experiment, which in fact are meadow experiments (grazings seems to be missing; also a major deficit). Such critical assessment would be extremely helpful to guide the installation of future LTFEs and to show the limitations in the conclusions that can be drawn from the existing LTFEs. | We will include grazing as example in the discussion of limitations of existing LTFEs. |
| Were lysimeter experiments included, which would allow assessing at least vertical water fluxes? Do long-term experiments with lysimeter exist at all in Germany? | No, lysimeter experiments were excluded because they were considered as an own category. Some reasons are that soils are often transferred and not undisturbed in lysimeter experiments and tillage has to be conducted by hand instead of machines, which can bias some results. Indeed, longterm lysimeter experiments exist in Germany as part of the Tereno network. We will clarify this. |
| Were experiments included that allow quantification of lateral processes (runoff, soil loss)? I could imagine that the measurements in Trier (Stehling and Schmidt 2017) or those by Jung and Brechtel (1980) qualify for LTFE. If they don't qualify, this would again illustrate a major deficit of present LTFEs. | Our response to the first review comments also holds here. In addition, we check again the mentioned resources to evaluate that. |
| In the discussion I missed a wider view. Do similar compilations also exist in other countries? Are the German LTFE experiments similar to what was done and is done in other countries? | We will include a section about the international situation. |
| Furthermore, the authors give the impression that they still focus on the old questions of LTFEs (mainly yield) that became boring. I had this impression for two reasons. First, little examples are given how LTFEs can be used in fascinating modern research on urgent questions. Second, using LTFEs in modern research applying new techniques requires access to the experiments. Hence it makes a big difference whether an experiment is still ongoing or not. However, this information is given nowhere. Second, it often requires | Information on whether an LTFE still exists or not can be found in the extensive data set, which can be found under the following DOI: http://doi.org/10.20387/BonaRes-3tr6-mg8r, 2019
We asked in a questionnaire whether there were any retain samples or not. This information is available for 40 LTFE. The relatively small number compared to the total number stopped us from integrating this information into the data set. But we can do that in a next version of the data set. |

| | |
|---|---|
| archived samples (as an example what can be done with modern techniques and archived samples, Köhler et al. 2012 comes to my mind but there are certainly more examples). This information, whether archived samples are available, should be included. Generally, I missed information about which data could be obtained from the LTFEs. | Information about which data have been collected in the experiments can also be found in the dataset ("research parameters"). |
| Most of my other remarks are mainly editorial issues. The weakest part in this respect is the table in the Appendix, which is most important because it resolves the LTFEs and thus allows access (see below). | We could imagine to provide the whole dataset, which is very extensive, as supplemental material instead of the Appendix. |
| 12: add "during the growing season"; I would even change the abbreviation to CWBg because usually an entire year is considered in a CWB. I was very surprised when suddenly somewhere in the manuscript the information 'growing season' popped up | We will do that. |
| 13: Müncheberger Soil Quality Rating seems to be a combination of German and English. Shouldn't it be 'Müncheberg'? | You are right, we will change accordingly. |
| 35: I welcome this definition of the control that is certainly better than the often used but wrong assignment of the strongest and most unrealistic intervention as control, namely the long-term nutrient removal. However, I did not find this definition to be used later in the manuscript. | Yes, we used this term only to give an example on how LTFE could be analysed collectively. We are going to write this part more detailed, also due to the comments of Referee #3. |
| 46: Bai et al. | We will change accordingly. |
| 116: Not clear how PET was derived. Was it taken from DWD? Is it Haude? | The PET was already included in the DWD data of CWB. |
| 126: This is strange. Later only 6 classes of the MSQR are used, not 102. I wonder whether different properties like soil structure, wetness, relief, contaminations can be combined in one indicator of six classes. This may be possible for one specific target like yield but will fail for most other targets or require other classes. Is a better resolution than these six classes possible? | The soil qualita rating is is performed on an ordinal scale of 0-102 and clustered into six quality classes. We will add this information to clarify. |
| 128: I guess this should read 'available water capacity' | The source says 'profile available water', just as Mueller 2010 |
| 130: What is unsuitable? This always requires the definition of a target. | We cited the source correctly, but we can add "for crop production" here. |
| 139: This leads to the question: Were lysimeter experiments included? If not, why not? | See above |
| 155: The title does not have this restriction; also the Abstract does not. I wonder why it suddenly pops up in the results. I also wonder how this is defined (what is bioeconomy?) and whether these experiments really aim at sustainable soil | Most LTFE were originally implemented for agronomic purposes. Accordingly and particularly for grassland LTFE, most research questions are agronomic in nature and not closely related to the soil. In this paper, we |

| | |
|---|---|
| use. They exclude many things that make soil use unsustainable (erosion, compaction) and hence are unsuitable to test sustainability (in this general sense). I also wonder even more why the criterion sustainability excludes some grassland experiments. This is contrary to what I would expect. | intended to reveal the value of LTFE for soil related questions. We therefore only included those LTFE in our study, for which soil data are existing. We will state this more clearly. Besides that we will point to the deficits in LTFE setup as mentioned above (erosion, compaction, grazing). |
| 160: Establishment was in the past. Hence past tense would be appropriate. The question of correct tense is rather difficult to answer given that 30% of the experiments have come to an end already and others will come to an end in the future, I wonder whether the mostly used present tense is justified. | Ok, we consider past tense. |
| 171-172: One sentence is usually not a paragraph. Furthermore, temporal aspects were treated in the first paragraph of the results. I suggest moving this sentence. | Ok, good idea. |
| 173: sentences usually do not start with a number; this also applies in other cases (e.g. L. 181, 184). | Ok, we will write out the numbers with letters; I think that will be correcht. |
| 178 : Move opening parenthesis | Yes, thank you |
| 208-209: This should be moved to the M & M section; this is the first time that growing period is mentioned although CWB appeared already several times. Furthermore, it would be good to explain the rationale behind this decision than let the reader speculate | Ok |
| 266-269: I would reverse the argument. In my view the critique by Franko is well justified and shows that 6 classes of the MSQR are insufficient. I do not suggest to include an assessment of the complexity of soil parameters but it is also not justified to say that the LTFEs are representative regarding soils just because they match the rather coarse and restricted (to yield) MSQR criterion. | We agree. We intended to say which CWB/MSQR combinations are less well represented in the existing LTFE having biomass production suitability in mind. For specific questions such as the representation of C-dynamics in simulation models other requirements to long term information exist. We will clarify the part. Furthermore most likely we are going to include in addition to MSQR and CWB an assessment of the distribution of LTFE according to clay content with clay data from JRC. |
| References: The format varies among references. Please homogenize | Ok |
| Fig. 2: The pie charts are an attempt to illustrate the manuscript. However, they do a poor job. They require a legend, which is difficult to read (because font size is smaller than that of ordinary text) and contain information that is better suited for a table or even could be given as plain test. For Fig. 2 a, a density graph would be more appropriate | Ok, we put this information into a table respectively a density graph. |
| Fig. 3: A graph usually has not a title but a caption. The colors are impossible to distinguish | We are going to change the colours respectively to change the whole figure. |

| | |
|---|---|
| Are they necessary? Can they be simplified? Wouldn't the year when an LTFE was closed be equally interesting? | |
| Table 1: It is not clear whether 'organic fertilization' also includes straw and compost (there is not an equivalent 'Mineral fertilization'). Furthermore, why are green manure, compost and sludge mentioned, but not the main type of organic manure? This classification appears inconsistent. It surprises me that only two of the grassland experiments have organic fertilizer although grassland use unavoidably produces manure. Have all except for two experiments used an unrealistic design that does not allow application of the results to typical situations? Better call 'plant protection' 'crop protection' | We will improve the table accordingly. |
| Fig. 4: same remark as Fig. 2 | Ok |
| Table 2 + 3: 'vegetation period' should not be in the column head but in the caption. Also the lines separating groups of variables are not consistent (why are CWB class and range separated by a line? Isn't the unit for CWB mm/yr? | We will change the tables accordingly. |
| Fig. 5: Here four classes of LTFE are sufficient. Why does Fig. 3 require eight classes (that cannot be read anyhow)? LTFE should not be repeated five times in the legend. It is not necessary at all. CWB is in mm/yr | For the map we simplified the classes to avoid complexity. We are going to simplify figure 3 also. We skip LTFE from the legend. We change the unit of CWB. |
| Fig. 6: Delete LTFE | Ok |
| Fig. 7: column widths could be much smaller while larger row heights would allow a larger font size. Presently the numbers hardly can be read. It is not necessary repeating 'MSQR class' six times. Better use a larger font size. The colors of the legend should agree with the colors in the graph. | We will change the figure. Referee #1 also commented on this figure and suggested points or lines. |
| Table A 1: This is likely the most important table because it allows access to the LTFEs. However, it is rather inconsistent and difficult to read. E.g., the IDs cannot be read; some institutions got abbreviations (why?) others not; some places are mentioned, others not (why?). Mentioning the main institution may be fine in hierarchical organizations but this is clearly insufficient for big universities. Whom should one ask there? I suggest replacing the information in column 3 by a number and the place and resolving the number below the table by reporting the full addresses. This would also create room for the other columns. Furthermore, I see no reason why umlauts are | We will change the table accordingly. |

| replaced. This is poor technology of the past century and again a waste of space. | |
|---|---|

---

## Author Comment (AC2) · 27 Jul 2020

**Author Response to Anonymous Referee #3**

| Review comment | Author response |
|---|---|
| The Material and Methods chapter explains how the geospatial analysis is done and also the classification criteria for the LTFEs. However, there is no information on how the experimental design should be analyzed as stated as one of the two main objectives of this study. | We are going to write a section about the analysis of LTFE. There the analysis of individual LTFE shall be described as well as the analysis of several LTFE with similar treatments in the form of a meta-analysis. However, in our opinion, this section then will fit better in the introductory part. |
| Do statistical methods come to use? Which ones? The pure assignment of LTFEs to four different classes without further statistical analyses (e.g. various types of discriminant analysis, contingency and cross tabulation, factor analysis) is not very appealing. The same holds true for the analysis of the data for climate (CWB) and soil fertility (MSQR) given as number of cases and percentage of share of classes (tables 2 and 3). | It is important to stress that our database comprises a complete respository of all LTFE with a duration of more than 20 years conducted in Germany. As such, our database constitutes a complete enumeration of the whole population of LTFE in Germany. Due to the complete enumeration, we believe that descriptive statistics (cross-tabulations, contingency tables) provide the best means of analysing our data. We will write the two used methods down in the Material and Methods section. Methods of statistical inference, such as chi-squared tests for contingency tables, for example, are unecessary, precisely because of the complete enumeration. Such tests would only be helpful, if a random sample of LTFE were available out of a larger population. But such is not the structure of our data. |
| | The reviewer also suggests two multivariate methods, i.e., as factor analysis and discriminant analysis. Both methods would potentially use a large number of environmental covariates characterizing the LTFE. By contrast, our hypotheses relate to two clearly defined covariates that span a two-way classification, i.e. Müncheberg Soil Quality Rating and Climatic Water Balance. Moreover, we believe the two suggested multivariate techniques do not really match our objectives. The purpose of discriminant analysis it to provide a model-based decision rule that allows allocating new samples to known groups of units (LTFE in our case). This kind of application is clearly not what we need, as we already have a classification of all LTFE in our database. Moreover, there are no new LTFE to be classified. As regards factor-analysis, this is largely an exploratory method for a larger number of variates that allows exploring possible grouping in multivariate space. Again, this does not meet our needs; we already have the classification in hand that we are analysing, |

| | and this is based on just two well-defined covariates. |
| --- | --- |
| | Further on, we are going to include a further author, who is an expert i.a. in spatial methods for field trials, design of comparative experiments, and network meta-analysis. |
| (There are) five (classes of LTFE) in table 1 and eight in figure 3? | This is due to the fact that multiple nominations were possible in Table 1 and we wanted to deal with each individual LTFE in Figure 3. We will redesign Figure 3, also according to the comments of the other referees, and clarify what exactly we want to represent. |
| I am convinced that the manuscript would greatly benefit from a profound statistical analysis and that this would allow (i) a critical discussion of the value of the data that exist so far and (ii) to conclude how such laborious and expensive experiments could be designed in future. | See response to the 2nd statement. |
| A purely qualitative, merely descriptive analysis has certainly been carried out to a sufficient extent in the large number of papers already published on this subject, most of them mentioned generously. | Although various compilations of LTFE in Germany exist, this paper is new in the aspect, that it provides a carefully developed example on how a large number of long-term field experiments can be comprehensively characterized with meta-information. In addition, the geospatial analysis of LTFE sites is new. We will clarify this and further reveal the added value of knowledge gained in this paper. |
| A discussion of the results including international literature and experiences of long-term experiments, e.g. from England, China or the US, is missing to a large extent. I recommend that the discussion be significantly revised and expanded in these points. | We will include international literature in the discussion. |
| Appropriate quantitative methods for the analysis of the experimental design and the spatial distribution of the experiments with regard to climate and soil fertility should be added. | What is meant by "experimental design" here? We have chosen a descriptive approach to classify the total population of LTFE in Germany. We believe that contingency and cross tabulation are stringent methods for this. If instead e.g. a factor analysis would have been chosen, that would be a completely different approach. |
| Line 49-55: the enumeration of the number of LTFEs published over the years by Körschens seems unnecessary in this way. If the details here are important I would recommend to present it as a table. And Line 83: after the explanations in the introduction regarding the work on the German LTFEs prepared by Koerschens et al., it seems | The numbers show, that our work was needed. We had the opportunity to carry out an extremely extensive search, which led to more than twice as many LTFE (205) being known as in Körschens' most extensive study (97). In addition, the setup of new LTFE with a planned duration of at least 20 years goes on and we have also recorded LTFE that were setup after |

| | |
|---|---|
| incomprehensible why a new literature study should be made here and would require a corresponding justification. This should also explain why the work of Koerschens et al. is obviously not adequate to follow the objectives of this study. | Körschen's publications. In addition, we included grassland experiments.
Also regarding the details to each of the experiments we provide much more information in our dataset (http://doi.org/10.20387/BonaRes-3tr6-mg8r). Although most of the details are not needed for the spatial analyses of this paper, the precise coordinates of the LTFE are needed and could only be found out through our extensive search. We will state that more clearly in the paper. |
| Lines 63-80: after the objectives of the work have been formulated in lines 61-63, the explanations given here seem like a description of material and methods. I recommend to shorten this part and to integrate it into the chapter Material and Methods. | Ok, we will shorten it and integrate it into Material and Methods. |
| Line 68: what is meant by research parameters? Please list. | By research parameter we mean everything that has ever been sampled and recorded in LTFE. Probably "measured parameters" is less misunderstanding. We will change that. An overview of the measured parameters known to us can be found on pages 9 to 11 of the fact sheet (Grosse, M., Heinrich, U., and Hierold, W.: Fact Sheet for the Description of Long-Term Field Experiments / Steckbrief zur Beschreibung von Dauerfeldversuchen, http://doi.org/10.20387/BonaRes-R56G-FGRW, 2019.). That is a very long list. Maybe we can simply refer to that? |
| Line 95: here, too, the technical justification for the selected research topics is missing. Especially with regard to the aspect of a meta-analysis of the research statements, which was prominently emphasized in the introduction, the research topics listed here appear incomplete. | We will insert a short explanation in line 95, why we clustered the LTFE according to these four themes.
Furterhon there may be a misunderstanding here. We did not promise a meta-analysis in the introduction. It is a (descriptive) analysis of the LTFE in Germany with regard to land use, research themes and farming systems (lines 61-62). We will add "descriptive" and skip "experimental design", which probably lead to the misunderstanding. |
| Lines 200-206: the description of the methodology belongs in the corresponding chapter and is superfluous here, as are lines 208 and 209. Similar mixtures of results and material and methods are also shown in the following chapters. I would recommend to check the results part and to concentrate all methodical information at the appropriate place. | We will enhance the structure. |
| Figure 1 does not seem necessary to me, the content is very simple and directly repeats the | We would like to leave Figure 1, because we believe it improves the readability of the paper. |

| | |
|---|---|
| statements in the text without a gain in information. | |
| The core statements in figure 3 could certainly be presented much more clearly. At the moment most of the space is taken up by the legend. It also seems unusual to me that the figure itself contains a headline ('Start of LTFE'). | We will enhance Figure 3, see above. |

---

## Author Comment (AC3) · 28 Jul 2020

Author Response to Anonymous Referee #1

| Review comment | Author response |
|---|---|
| For the international readership of SOIL it might be of limited interest, since all results are related to Germany without direct implications for outside Germany | This comment was already contradicted by reviewer 3. Indeed, the paper is exclusively about LTFE in Germany. But we expect the paper to be also interesting for an international readership because it provides a carefully developed example on how a large number of long-term field experiments can be comprehensively characterized with meta-information. On the other hand, the intersection of LTFE with spatial data is new and could also arise the interest of international readers, either with regard to the specific data usage of the German LTFE, or as inspiration for using their own LTFE. |
| More details need to be outlined how the access to data will be provided in the future. The one sentence in l. 68 ("There is a focus on reseach data from LTFEs") is not enough | We see this paper as a kind of vision or as motivation to make the LTFE data freely available. We expect that the comprehensive overview of meta-information will trigger motivation of LTFE holders to share their data for re-use and scientific cooperation. It facilitates the direct (bilateral) cooperation between interested scientists and LTFE holders for co-authorship. Nevertheless, we will go more into detail in the manuscript how to convince LTFE operators to enter the data in the BonaRes database. |
| Maybe more abundant soil data, such as texture or soil type, can be used for classification and the representativeness analysis | We are going to conduct a further analysis with texture or soil type data. |
| l. 6: Soil monitoring of climate impact can be performed much more cost efficient on permanent sampling sites (such as "Bodendauerbeobachtung"). Since LTFEs do not represent real practice field sites they might miss some trends that can only be monitored at farmers' field sites. The value of LTFEs is to provide data on management impacts (under changing climate). | Both methods and programs have their specific goals and advantages. While soil monitoring sites (BDF) show soil changes during normal management, LTFE follow an experimental design. The strengths of the collective analysis of LTFE therefore is the analysis of LTFE with similar treatments in the form of a meta-analysis. However, if the LTFE data are available anyway, it is also conceivable to use LTFE in a similar way to BDF by evaluating the "conventional fertilization and tillage" treatments collectively that are existing in most LTFE. |
| L 16: The representation and distribution of management options in the LTFEs is missing as a result in the abstract. Since this is the main aim of LTFEs it would be worth to include one | We are going to include that in the abstract. |

| | |
|---|---|
| or two sentences on how management treatments are covered in LTFEs in Germany. | |
| l. 28: In agriculture, plant nutrition is linked to fertilisation. Thus, these are not two but one and the same aspect. | We are going to express this more clearly |
| l. 39: The definition of "control treatments" is not clear. Is the control treatment defined by each LTFE or does it depend on the study? Customary or common management practices are changing over time e.g. the fraction of reduced tillage or fertilisation type and amount. Is the control treatment than also changing over time? | The definition we gave here is for the purpose of defining 'control treatment' for our study. The second point is a fundamental problem for long time series of LTFE, since the management changes repeatedly over time. This must be considered in individual time series to see how strong the breaks are and whether or not these time series can then be used. |
| l. 45: Change "landscapes to "soil" since LTFEs does not comprise landscapes. | This refers to the collective analysis of LTFE. We will state that more clearly. |
| l. 99 and 102: Why 191? 94+87=181 | We will correct that. |
| l. 156: It is not comprehensible why many grassland LTFEs were excluded. This need to be explained and justified since grassland trials are under-represented in the compiled LTFE dataset. Above it is written that LTFEs are useful beyond the original scope or research theme. Here it is argued that the research theme of the grassland trials did not fit and were therefore excluded. | Most LTFE were originally implemented for agronomic purposes. Accordingly and particularly for grassland LTFE, most research questions are agronomic in nature and not closely related to the soil. In this paper, we intended to reveal the value of LTFE for soil related questions. We therefore only included those LTFE in our study, for which soil data are existing. We will state this more clearly. |
| l. 192: What is a technical college? A university of applied sciences? | Sorry. Yes, university of applied sciences. We will correct that. |
| l. 200-206: This section is redundant and repetition from above an can be removed. | We will do that. |
| l. 214-l. 223: For an international readership of the journal, it would be good to provide a map with the names of the regions mentioned here or include the names in Fig 5. | We are going to do the one or the other. |
| Fig. 3: The colours are not easy to distinguish, in particular that for tillage, fertilisation and crop rotation. | We are going to change the colours respectively change the whole figure according to the comments of the other reviews. |
| Fig 5 and 6: The dispersion of points from only single experimental sites with different experiments results in biased impressions, e.g. that the whole region of Halle is covered with LTFEs even though there might be only one single experimental site. I propose to either strongly reduce the dispersal of the points from one site or completely avoid them since this map aims at illustrating the spatial distribution and representativeness of LTFEs and one site with many trails mostly does not contribute in achieve a higher representativeness of soils and climate. | OK. We will revise these illustrations, combine the points per location and subject, and adjust the point size according to the number of LTFE at a location. |
| Fig 5: The map seems to be incomplete for German agricultural land (with is the reference for this study). Mostly grassland seem to be | We think CORINE is a good basis because CORINE is also available for Europe. It is raised according to the same rules within Europe, uses |

| | |
|---|---|
| missing, e.g. in the pre Alps, the Sauerland or in North-Western Germany. Readers expect that the class "other land" comprise only non-agricultural land. Maybe CORINE data are not appropriate but ATKIS Basis DLM data can be used. | a uniform legend and would therefore ensure connectivity. ATKIS is specific to Germany and is outside of Germany not relevant. CORINE provides data for a reference year. ATKIS has a permanent update cycle of 5 years. Each federal state does this on its own. Every year a fifth of every state is being photographed (aerial photos), preferably in spring, and updated on this basis. So there is not land use for one year but a mosaic of 5 years. For this reason and the fact that the aerial photos come from spring, the differentiation of arable and grassland is not so easy at ATKIS. For these reasons we would like to continue using CORINE. |
| Fig. 7: This illustration with boxes is unusual and thus difficult to read. Since the yaxis contains distinct values (no classes) a representation with points or lines would be more appropriate. | We will change the figure. Referee #2 also commented on this figure and suggested smaller column widths and larger row heights. |

---

## Author Response (AR1)

**Long-term Field Experiments in Germany: Classification and spatial Representation**

Meike Grosse[1], Wilfried Hierold[1], Marlen C. Ahlborn[1], Hans-Peter Piepho[2], Katharina Helming[1]

[1]Leibniz Centre for Agricultural Landscape Research (ZALF), Eberswalder Straße 84, Müncheberg, 15374, Germany
5   [2]Biostatistics Unit, Institute of Crop Science, University of Hohenheim, 70599 Stuttgart, Germany

*Correspondence to*: Meike Grosse (meike.grosse@zalf.de)

**Abstract.** The collective analysis of long-term field experiments (LTFE), here defined as agricultural experiments with a minumum duration of 20 years and research in the context of sustainable soil use and yield, can be used for detecting changes in soil properties and yield such as induced by climate change. However, information about existing LTFEs is
10   scattered, and the research data are not easily accessible. In this study, meta-information on LTFEs in Germany is compiled and their spatial representation is analysed. The study is conducted within the framework of the BonaRes project, which, inter alia, has established a central access point for LTFE information and research data. A total of 205 LTFEs is identified with a minimum duration of twenty years and research in the context of soil and yield 
[revised manuscript text omitted]

105 ~~Hierold, 2019). It contains information about name of the LTFE, exact location, holding institution, land use categorie, participation in existing networks, research theme, start (and maybe end) of the trial, and research parameters. Besides the focus of this project on the acquisition of metadata there is a focus on research data from LTFEs. The aims are to make LTFEs more visible, to enhance networking among LTFEs and to simplify common analyses of LTFEs. In compiling the dataset, special attention was focused on LTFEs with a minimum duration of 20 years. This age can be seen as a threshold~~
110

115 ~~representativeness of LTFEs according to these site sizes was assessed. LTFEs are classified according to their land use and their research themes to simplify the identification of similar experiments. The identification of suitable LTFEs in similar (or different) landscapes shall be enhanced. Therefore, a table with the IDs of all experiments, their thematic classification, their CWB class and their MSQR class is provided in the attachment. More details for each LTFE can be identified in the published dataset (Grosse and Hierold, 2019) through the ID of the LTFE. Thus, cooperation with LTFE holders can be~~
120

**2 Material and Methods**

A combination of three methods was applied: a literature review to identify LTFEs in Germany, a fact sheet-based addition of information to the identified LTFEs, and a geospatial analysis employing the CWBg and the MSQR (Figure 1).

An extensive literature review was conducted to identify LTFEs. The search  terms were 'long-term field experiment',
125 'long-term experiment', 'long-term field trial', and 'long-term trial', as well as the German items 'Dauerfeldversuch', 'Dauerdüngungsversuch', 'Dauerversuch', 'Langzeitfeldversuch' and 'Langzeitversuch'. Sources were scientific papers as well as other articles, books, trial guides and websites. The focus was on the exact position of the LTFE and the following

metadata: name of the LTFE, website (if available), institution, land use category, participation in existing networks, research theme, size of the LTFE area, number of plots, size of the plots, crop rotation, start (and maybe end) of the trial, research measured parameters, and trial setup including factors, treatments and randomization. For the coordination and simplification of the trial description, the BonaRes Fact Sheet was established, which asks for all relevant trial information (Grosse et al., 2019). It was sent to the trial holders, and the fact sheet was completed for 40 trials. Trial holders also delivered important information as personal communication.

In compiling the dataset, special attention was paid to LTFEs with a minimum duration of 20 years. This age can be seen as a threshold for the identification of long-term trends. Attention was given to LTFEs in the context of soil research, i.e., the objects of research should at least include soil properties and yield as an important soil function. The setup of each trial should allow for statistical analyses, i.e., have clearly defined treatment factors, replications and as much as possible a static design. Lysimeter experiments were excluded because they were considered as an own category. Some reasons for this exclusion are that soils are often transferred and not undisturbed in lysimeter experiments and tillage has to be conducted by hand instead of machines, which can bias some results. Indeed, longterm lysimeter experiments exist in Germany as part of the TERENO network (TERENO, 2020).

The LTFEs were classified according to their research themes to simplify the identification of similar experiments. The field crops LTFEs could best be grouped into four clusters: fertilization, tillage, crop rotation, other. The fourth cluster "other" entails all themes that could not be grouped into the first three and appeared only in a few (maximum five) LTFE cases, so that a separate group was not justified. following research themes were selected: fertilization, tillage, crop rotation or other research themes. LTFEs were considered to belong to one group if one factor was fertilization, tillage, crop rotation, or another theme. 
[revised manuscript text omitted]

In the following analyses, the number of LTFEs is compared to the proportion of classes of CWB and MSQR, separately,
305 according to their research topics (fertilization, tillage, crop rotation). Fertilization experiments are subdivided into field crops (including two vegetable experiments) or grassland experiments. In tillage and crop rotation experiments, no grassland experiments exist. While the CWB is available for the whole territory and can be evaluated separately for arable land and grassland, the MSQR soil quality is available only for arable land.

The total numbers of experiments in these analyses are 158 fertilization experiments (124 field crops and 34 grassland
310 experiments), 38 tillage experiments and 32 crop rotation experiments (multiple nominations possible).

**3.2.1 Geospatial Analysis of LTFEs in Relation to the Climatic Water Balance of the growing season (CWBg) Distribution**

For the analysis, the CWB of the vegetation period (1 May to 31 October) was used according to Survey Guideline KA5 (Ad hoc AG Boden, 2005). 
[revised manuscript text omitted]

| 1 | <-150 | 599 247 | 14 | 6 | 18 |
| 2 | -150 - <-50 | 792 064 | 18 | 3 | 9 |
| 3 | -50 - <50 | 1 420 319 | 33 | 15 | 44 |
| 4 | 50 - <150 | 398 496 | 9 | 7 | 21 |
| 5 | 150 - <300 | 1 009 952 | 23 | 3 | 9 |
| 6 | 300 - <500 | 137 968 | 3 | 0 | 0 |
| 7 | >500 | 0 | 0 | 0 | 0 |

585

[Figure]

Coordinate System: ETRS 1989 Transverse Mercator
Projection: Transverse Mercator
Date: ETRS 1989

Data: CORINE (CLC) 2018;
Grosse & Hierold 2019

| 0 | 50 | 100 | 200 |

Kilometers

**Long-term field experiments (LTFE)**

☐ LTFE focused on fertilization (n = 124)

● LTFE focused on tillage (n = 38)

⬠ LTFE grassland (n = 34)

★ LTFE focused on crop rotation (n = 32)

**Climatic water balance [mm]**

<-150

-150 - <-50

-50 - <50

50 - <150

150 - <300

300 - <500

>=500

other land use area

[Figure]

| Long-term field experiments (LTFE) | Climatic water balance [mm/yr] |
| --- | --- |

Figure 54: Overview of the distribution of the different climatic water balance classes of the growing season and the different LTFE types in Germany. . The size of the symbols varies according to the amount of LTFEs at one place.

Table 4: Müncheberg Soil Quality Rating (MSQR) classification of arable land in Germany and the number or share of the different LTFE types in each MSQR class.

| MSQR | Agricultural area | | LTFEs total (arable land) (n=169) | | Fertilization LTFEs* (n=123) | | Tillage LTFEs* (n=38) | | Crop rotation LTFEs* (n=32) | |
|---|---|---|---|---|---|---|---|---|---|---|
| | area [ha] | share [%] | number | share [%] | number | share [%] | number | share [%] | number | share [%] |
| extremely low | 705 687 | 6 | 9 | 5 | 5 | 4 | 4 | 11 | 3 | 9 |
| very low | 2 149 584 | 17 | 29 | 17 | 22 | 18 | 5 | 13 | 5 | 16 |
| low | 2 656 535 | 21 | 18 | 11 | 13 | 11 | 3 | 8 | 1 | 3 |
| medium | 3 532 109 | 28 | 32 | 19 | 28 | 23 | 6 | 16 | 4 | 13 |
| high | 2 182 221 | 18 | 45 | 27 | 28 | 23 | 13 | 34 | 11 | 34 |
| very high | 1 181 237 | 10 | 36 | 21 | 27 | 22 | 7 | 18 | 8 | 25 |

*multiple nominations possible

[Figure]

[Figure]

600    Figure 65: Overview of the distribution of the different Müncheberger Soil Quality Rating classes and the different LTFE

types in Germany. The size of the symbols varies according to

the amount of LTFEs at one place.

[Figure]

605

[Figure]

Figure 76: Share of arable area and LTFEs in every climatic water balance – Müncheber Soil Quality Rating combination.

610

Table 5: Clay content classification according to ESDAC (2020) of arable land in Germany and the number or share of the different LTFE types in each clay content class.

| Clay content class | Range [%] | Agricultural area (arable) | | LTFEs total (arable land) (n=169) | | Fertilization LTFEs* (n=124) | | Tillage LTFEs* (n=38) | | Crop rotation LTFEs* (n=32) | |
|---|---|---|---|---|---|---|---|---|---|---|---|
| | | area [ha] | share [%] | number | share [%] | number | share [%] | number | share [%] | number | share [%] |
| 1 | 0 to 5 | 1 748 393 | 14 | 25 | 15 | 19 | 15 | 6 | 16 | 3 | 9 |
| 2 | 6 to 10 | 2 404 798 | 19 | 24 | 14 | 19 | 15 | 6 | 16 | 3 | 9 |
| 3 | 11 to 16 | 2 265 517 | 18 | 29 | 17 | 20 | 16 | 5 | 13 | 4 | 13 |
| 4 | 17 to 19 | 1 523 493 | 12 | 42 | 25 | 37 | 30 | 6 | 16 | 6 | 19 |
| 5 | 20 to 21 | 1 179 602 | 9 | 15 | 9 | 12 | 10 | 2 | 5 | 8 | 25 |
| 6 | 22 to 24 | 1 553 463 | 12 | 20 | 12 | 11 | 9 | 5 | 13 | 6 | 19 |
| 7 | 25 to 27 | 1 097 725 | 9 | 4 | 2 | 1 | 1 | 3 | 8 | 1 | 3 |
| 8 | 28 to 98 | 1 082 066 | 8 | 10 | 6 | 5 | 4 | 5 | 13 | 1 | 3 |

Table 6: Clay content classification according to ESDAC (2020) of agricultural area used for grassland in Germany and the number or share of the LTFEs on grassland in each clay content class.

615

| Clay content class | Range [%] | Agricultural area (grassland) | | Grassland LTFEs (n=34) | |
|---|---|---|---|---|---|
| | | area [ha] | share [%] | number | share [%] |
| 1 | 0 to 5 | 715 137 | 11 | 3 | 9 |
| 2 | 6 to 10 | 941 166 | 15 | 5 | 15 |
| 3 | 11 to 16 | 952 126 | 15 | 0 | 0 |
| 4 | 17 to 19 | 821 432 | 13 | 4 | 12 |
| 5 | 20 to 21 | 710 826 | 11 | 6 | 18 |
| 6 | 22 to 24 | 978 366 | 15 | 5 | 15 |
| 7 | 25 to 27 | 651 066 | 10 | 8 | 24 |
| 8 | 28 to 98 | 639 561 | 10 | 3 | 9 |

620

**Appendix**

Table A 1: IDs of all long-term field experiments, their original name, their place, their CWBg class (1 May to 31 October), their MSQR class, and their thematic classification. The institutional address is indicated by a number and given below the table. More details about the LTFEs can be found in the complete dataset (Grosse & Hierold, 2019).

[revised manuscript text omitted]

**Author Response to Anonymous Referee #1**
(Author Responses to Anonymous Referees #2 and #3 see below)

| Review comment | Author response |
| --- | --- |
| For the international readership of SOIL it might be of limited interest, since all results are related to Germany without direct implications for outside Germany | This comment was already contradicted by reviewer 3. Indeed, the paper is exclusively about LTFE in Germany. But we expect the paper to be also interesting for an international readership because it provides a carefully developed example on how a large number of long-term field experiments can be comprehensively characterized with meta-information. On the other hand, the intersection of LTFE with spatial data is new and could also arise the interest of international readers, either with regard to the specific data usage of the German LTFE, or as inspiration for using their own LTFE. |
| More details need to be outlined how the access to data will be provided in the future. The one sentence in l. 68 ("There is a focus on reseach data from LTFEs") is not enough | We see this paper as a kind of vision or as motivation to make the LTFE data freely available. We expect that the comprehensive overview of meta-information will trigger motivation of LTFE holders to share their data for re-use and scientific cooperation. It facilitates the direct (bilateral) cooperation between interested scientists and LTFE holders for co-authorship. We entered some more details about the common database. |
| Maybe more abundant soil data, such as texture or soil type, can be used for classification and the representativeness analysis | We conducted a further analysis with clay data. |
| l. 6: Soil monitoring of climate impact can be performed much more cost efficient on permanent sampling sites (such as "Bodendauerbeobachtung"). Since LTFEs do not represent real practice field sites they might miss some trends that can only be monitored at farmers' field sites. The value of LTFEs is to provide data on management impacts (under changing climate). | We included a section about Bodendauerbeobachtungsflächen (lines 52 to 55). |
| L 16: The representation and distribution of management options in the LTFEs is missing as a result in the abstract. Since this is the main aim of LTFEs it would be worth to include one or two sentences on how management treatments are covered in LTFEs in Germany. | We are included that in the abstract. |
| l. 28: In agriculture, plant nutrition is linked to fertilisation. Thus, these are not two but one and the same aspect. | We expressed this more clearly |

| | |
|---|---|
| l. 39: The definition of "control treatments" is not clear. Is the control treatment defined by each LTFE or does it depend on the study? Customary or common management practices are changing over time e.g. the fraction of reduced tillage or fertilisation type and amount. Is the control treatment than also changing over time? | The definition we gave here is for the purpose of defining 'control treatment' for our study. The second point is a fundamental problem for long time series of LTFE, since the management changes repeatedly over time. This must be considered in individual time series to see how strong the breaks are and whether or not these time series can then be used. |
| l. 45: Change "landscapes to "soil" since LTFEs does not comprise landscapes. | We changed that. |
| l. 99 and 102: Why 191? 94+87=181 | We corrected that. |
| l. 156: It is not comprehensible why many grassland LTFEs were excluded. This need to be explained and justified since grassland trials are under-represented in the compiled LTFE dataset. Above it is written that LTFEs are useful beyond the original scope or research theme. Here it is argued that the research theme of the grassland trials did not fit and were therefore excluded. | Most LTFE were originally implemented for agronomic purposes. Accordingly and particularly for grassland LTFE, most research questions are agronomic in nature and not closely related to the soil. In this paper, we intended to reveal the value of LTFE for soil related questions. We therefore only included those LTFE in our study, for which soil data are existing. |
| l. 192: What is a technical college? A university of applied sciences? | Sorry. Yes, university of applied sciences. We corrected that. |
| l. 200-206: This section is redundant and repetition from above an can be removed. | We removed that. |
| l. 214-l. 223: For an international readership of the journal, it would be good to provide a map with the names of the regions mentioned here or include the names in Fig 5. | We decided not to include the names in Fig. 5, as it would overload the figure. For an international readership we translated the names of the regions. We could provide a freely available map, but the names of the regions are in German language. |
| Fig. 3: The colours are not easy to distinguish, in particular that for tillage, fertilisation and crop rotation. | We changed the colours respectively changed the whole figure according to the comments of the other reviews. |
| Fig 5 and 6: The dispersion of points from only single experimental sites with different experiments results in biased impressions, e.g. that the whole region of Halle is covered with LTFEs even though there might be only one single experimental site. I propose to either strongly reduce the dispersal of the points from one site or completely avoid them since this map aims at illustrating the spatial distribution and representativeness of LTFEs and one site with many trails mostly does not contribute in achieve a higher representativeness of soils and climate. | We changed these illustrations, combined the points per location and subject, and adjusted the point size according to the number of LTFE at a location. |
| Fig 5: The map seems to be incomplete for German agricultural land (with is the reference for this study). Mostly grassland seem to be missing, e.g. in the pre Alps, the Sauerland or in North-Western Germany. Readers expect that the class "other land" comprise only non- | We think CORINE is a good basis because CORINE is also available for Europe. It is raised according to the same rules within Europe, uses a uniform legend and would therefore ensure connectivity. ATKIS is specific to Germany and is outside of Germany not relevant. CORINE |

| | |
|---|---|
| agricultural land. Maybe CORINE data are not appropriate but ATKIS Basis DLM data can be used. | provides data for a reference year. ATKIS has a permanent update cycle of 5 years. Each federal state does this on its own. Every year a fifth of every state is being photographed (aerial photos), preferably in spring, and updated on this basis. So there is not land use for one year but a mosaic of 5 years. For this reason and the fact that the aerial photos come from spring, the differentiation of arable and grassland is not so easy at ATKIS. For these reasons we would like to continue using CORINE. |
| Fig. 7: This illustration with boxes is unusual and thus difficult to read. Since the yaxis contains distinct values (no classes) a representation with points or lines would be more appropriate. | We changed the figure. Referee #2 also commented on this figure and suggested smaller column widths and larger row heights. |

**Author Response to Anonymous Referee #2**

| Review comment | Author response |
|---|---|
| All LTFE are situated in flat areas (a data evaluation in this respect would be nice and not too difficult to do). This means that they exclude major lateral processes (interflow, surface runoff) and differ largely from typical agricultural fields.
This deficit may be especially pronounced for grassland experiments because grassland either occupies lowland areas that are too wet for arable use or areas that are too steep. | Indeed, lateral processes are typically not analysed in LTFE and they are not designed for such questions. Different design such as the 'Wishmeyer plots' are implemented for erosion studies. We wrote a section explicitly about deficits in the setup of LTFE (lines 283-285). |
| For grassland experiment, which in fact are meadow experiments (grazings seems to be missing; also a major deficit). Such critical assessment would be extremely helpful to guide the installation of future LTFEs and to show the limitations in the conclusions that can be drawn from the existing LTFEs. | We included grazing as example in the discussion of limitations of existing LTFEs. |
| Were lysimeter experiments included, which would allow assessing at least vertical water fluxes? Do long-term experiments with lysimeter exist at all in Germany? | We included two sentences about lysimeter experiments in lines 137-140. |
| Were experiments included that allow quantification of lateral processes (runoff, soil loss)? I could imagine that the measurements in Trier (Stehling and Schmidt 2017) or those by Jung and Brechtel (1980) qualify for LTFE. If they don't qualify, this would again illustrate a major deficit of present LTFEs. | Our response to the first review comments also holds here. |
| In the discussion I missed a wider view. Do similar compilations also exist in other countries? Are the German LTFE experiments similar to what was done and is done in other countries? | We included a section about the international situation (lines 272-278). |
| Furthermore, the authors give the impression that they still focus on the old questions of LTFEs (mainly yield) that became boring. I had this impression for two reasons. First, little examples are given how LTFEs can be used in fascinating modern research on urgent questions. Second, using LTFEs in modern research applying new techniques requires access to the experiments. Hence it makes a big difference whether an experiment is still ongoing or not. However, this information is given nowhere. Second, it often requires archived samples (as an example what can be done with modern techniques and archived | Information on whether an LTFE still exists or not can be found in the extensive data set, which can be found under the following DOI: http://doi.org/10.20387/BonaRes-3tr6-mg8r, 2019
We included some information about archived samples (lines 261-263) and which data can be obtained from LTFEs (lines 254-260). |

| | |
|---|---|
| samples, Köhler et al. 2012 comes to my mind but there are certainly more examples). This information, whether archived samples are available, should be included. Generally, I missed information about which data could be obtained from the LTFEs. | |
| Most of my other remarks are mainly editorial issues. The weakest part in this respect is the table in the Appendix, which is most important because it resolves the LTFEs and thus allows access (see below). | We enhanced the table in the Appendix according to your suggestions. |
| 12: add "during the growing season"; I would even change the abbreviation to CWBg because usually an entire year is considered in a CWB. I was very surprised when suddenly somewhere in the manuscript the information 'growing season' popped up | We did that. |
| 13: Müncheberger Soil Quality Rating seems to be a combination of German and English. Shouldn't it be 'Müncheberg'? | We changed that accordingly. |
| 35: I welcome this definition of the control that is certainly better than the often used but wrong assignment of the strongest and most unrealistic intervention as control, namely the long-term nutrient removal. However, I did not find this definition to be used later in the manuscript. | Yes, we used this term only to give an example on how LTFE could be analysed collectively. We are wrote this part more detailed, also due to the comments of Referee #3. |
| 46: Bai et al. | We changed accordingly. |
| 116: Not clear how PET was derived. Was it taken from DWD? Is it Haude? | The PET was already included in the DWD data of CWB. |
| 126: This is strange. Later only 6 classes of the MSQR are used, not 102. I wonder whether different properties like soil structure, wetness, relief, contaminations can be combined in one indicator of six classes. This may be possible for one specific target like yield but will fail for most other targets or require other classes. Is a better resolution than these six classes possible? | The soil quality rating is is performed on an ordinal scale of 0-102 and clustered into six quality classes. We added this information to clarify. |
| 128: I guess this should read 'available water capacity' | The source says 'profile available water', just as Mueller 2010 |
| 130: What is unsuitable? This always requires the definition of a target. | We cited the source correctly, but we added "for crop production" here. |
| 139: This leads to the question: Were lysimeter experiments included? If not, why not? | See above |
| 155: The title does not have this restriction; also the Abstract does not. I wonder why it suddenly pops up in the results. I also wonder how this is defined (what is bioeconomy?) and whether these experiments really aim at sustainable soil use. They exclude many things that make soil use unsustainable (erosion, compaction) and | We included that in the abstract and avoided the term "bioeconomy". |

| | |
|---|---|
| hence are unsuitable to test sustainability (in this general sense). I also wonder even more why the criterion sustainability excludes some grassland experiments. This is contrary to what I would expect. | |
| 160: Establishment was in the past. Hence past tense would be appropriate. The question of correct tense is rather difficult to answer given that 30% of the experiments have come to an end already and others will come to an end in the future, I wonder whether the mostly used present tense is justified. | We changed to past tense. |
| 171-172: One sentence is usually not a paragraph. Furthermore, temporal aspects were treated in the first paragraph of the results. I suggest moving this sentence. | We moved the sentence. |
| 173: sentences usually do not start with a number; this also applies in other cases (e.g. L. 181, 184). | We wrote out the numbers with letters. |
| 178 : Move opening parenthesis | done |
| 208-209: This should be moved to the M & M section; this is the first time that growing period is mentioned although CWB appeared already several times. Furthermore, it would be good to explain the rationale behind this decision than let the reader speculate | We moved the sentence and explained, why we chose CWB of the growing season. |
| 266-269: I would reverse the argument. In my view the critique by Franko is well justified and shows that 6 classes of the MSQR are insufficient. I do not suggest to include an assessment of the complexity of soil parameters but it is also not justified to say that the LTFEs are representative regarding soils just because they match the rather coarse and restricted (to yield) MSQR criterion. | We agree. We intended to say which CWB/MSQR combinations are less well represented in the existing LTFE having biomass production suitability in mind. For specific questions such as the representation of C-dynamics in simulation models other requirements to long term information exist. We included in addition to MSQR and CWB an assessment of the distribution of LTFE according to clay content with clay data from ESDAC. |
| References: The format varies among references. Please homogenize | We homogenized the references. |
| Fig. 2: The pie charts are an attempt to illustrate the manuscript. However, they do a poor job. They require a legend, which is difficult to read (because font size is smaller than that of ordinary text) and contain information that is better suited for a table or even could be given as plain test. For Fig. 2 a, a density graph would be more appropriate | We put this information into a bar chart respectively plain text. |
| Fig. 3: A graph usually has not a title but a caption. The colors are impossible to distinguish Are they necessary? Can they be simplified? Wouldn't the year when an LTFE was closed be equally interesting? | We changed the whole figure. |
| Table 1: It is not clear whether 'organic | We improved the table accordingly. |

| | |
|---|---|
| fertilization' also includes straw and compost (there is not an equivalent 'Mineral fertilization'). Furthermore, why are green manure, compost and sludge mentioned, but not the main type of organic manure? This classification appears inconsistent. It surprises me that only two of the grassland experiments have organic fertilizer although grassland use unavoidably produces manure. Have all except for two experiments used an unrealistic design that does not allow application of the results to typical situations? Better call 'plant protection' 'crop protection' | |
| Fig. 4: same remark as Fig. 2 | done |
| Table 2 + 3: 'vegetation period' should not be in the column head but in the caption. Also the lines separating groups of variables are not consistent (why are CWB class and range separated by a line? Isn't the unit for CWB mm/yr? | We changed the tables accordingly. |
| Fig. 5: Here four classes of LTFE are sufficient. Why does Fig. 3 require eight classes (that cannot be read anyhow)? LTFE should not be repeated five times in the legend. It is not necessary at all. CWB is in mm/yr | For the map we simplified the classes to avoid complexity. We simplified figure 3 also. We skipped LTFE from the legend. We changed the unit of CWB. |
| Fig. 6: Delete LTFE | done |
| Fig. 7: column widths could be much smaller while larger row heights would allow a larger font size. Presently the numbers hardly can be read. It is not necessary repeating 'MSQR class' six times. Better use a larger font size. The colors of the legend should agree with the colors in the graph. | We changed the figure. |
| Table A 1: This is likely the most important table because it allows access to the LTFEs. However, it is rather inconsistent and difficult to read. E.g., the IDs cannot be read; some institutions got abbreviations (why?) others not; some places are mentioned, others not (why?). Mentioning the main institution may be fine in hierarchical organizations but this is clearly insufficient for big universities. Whom should one ask there? I suggest replacing the information in column 3 by a number and the place and resolving the number below the table by reporting the full addresses. This would also create room for the other columns. Furthermore, I see no reason why umlauts are replaced. This is poor technology of the past century and again a waste of space. | We changed the table accordingly. |

**Author Response to Anonymous Referee #3**

| Review comment | Author response |
|---|---|
| The Material and Methods chapter explains how the geospatial analysis is done and also the classification criteria for the LTFEs. However, there is no information on how the experimental design should be analyzed as stated as one of the two main objectives of this study. | We wrote a section about the analysis of LTFE. This section is inserted in lines 63 to 78. |
| Do statistical methods come to use? Which ones? The pure assignment of LTFEs to four different classes without further statistical analyses (e.g. various types of discriminant analysis, contingency and cross tabulation, factor analysis) is not very appealing. The same holds true for the analysis of the data for climate (CWB) and soil fertility (MSQR) given as number of cases and percentage of share of classes (tables 2 and 3). | It is important to stress that our database comprises a complete repository of all LTFE with a duration of more than 20 years conducted in Germany. As such, our database constitutes a complete enumeration of the whole population of LTFE in Germany. Due to the complete enumeration, we believe that descriptive statistics (cross-tabulations, contingency tables) provide the best means of analysing our data. In line 148 the two used methods are written down. Methods of statistical inference, such as chi-squared tests for contingency tables, for example, are unecessary, precisely because of the complete enumeration. Such tests would only be helpful, if a random sample of LTFE were available out of a larger population. But such is not the structure of our data.

The reviewer also suggests two multivariate methods, i.e., as factor analysis and discriminant analysis. Both methods would potentially use a large number of environmental covariates characterizing the LTFE. By contrast, our hypotheses relate to two clearly defined covariates that span a two-way classification, i.e. Müncheberg Soil Quality Rating and Climatic Water Balance. Moreover, we believe the two suggested multivariate techniques do not really match our objectives. The purpose of discriminant analysis it to provide a model-based decision rule that allows allocating new samples to known groups of units (LTFE in our case). This kind of application is clearly not what we need, as we already have a classification of all LTFE in our database. Moreover, there are no new LTFE to be classified. As regards factor-analysis, this is largely an exploratory method for a larger number of variates that allows exploring |

| | possible grouping in multivariate space. Again, this does not meet our needs; we already have the classification in hand that we are analysing, and this is based on just two well-defined covariates. |
|---|---|
| | Further on, we included Hans-Peter Piepho as a further author, who is an expert i.a. in spatial methods for field trials, design of comparative experiments, and network meta-analysis. |
| (There are) five (classes of LTFE) in table 1 and eight in figure 3? | We changed figure 3 so that it has also five classes of LTFE (with multiple nominations). |
| I am convinced that the manuscript would greatly benefit from a profound statistical analysis and that this would allow (i) a critical discussion of the value of the data that exist so far and (ii) to conclude how such laborious and expensive experiments could be designed in future. | See response to the 2nd statement. |
| A purely qualitative, merely descriptive analysis has certainly been carried out to a sufficient extent in the large number of papers already published on this subject, most of them mentioned generously. | Although various compilations of LTFE in Germany exist, this paper is new in the aspect, that it provides a carefully developed example on how a large number of long-term field experiments can be comprehensively characterized with meta-information. In addition, the geospatial analysis of LTFE sites is new. |
| A discussion of the results including international literature and experiences of long-term experiments, e.g. from England, China or the US, is missing to a large extent. I recommend that the discussion be significantly revised and expanded in these points. | We included a section about international LTFE in the lines 259 to 265. |
| Appropriate quantitative methods for the analysis of the experimental design and the spatial distribution of the experiments with regard to climate and soil fertility should be added. | What is meant by "experimental design" here? We have chosen a descriptive approach to classify the total population of LTFE in Germany. We believe that contingency and cross tabulation are stringent methods for this. If instead e.g. a factor analysis would have been chosen, that would be a completely different approach. |
| Line 49-55: the enumeration of the number of LTFEs published over the years by Körschens seems unnecessary in this way. If the details here are important I would recommend to present it as a table.
And
Line 83: after the explanations in the introduction regarding the work on the German LTFEs prepared by Koerschens et al., it seems incomprehensible why a new literature study should be made here and would require a | The numbers show, that our work was needed. We had the opportunity to carry out an extremely extensive search, which led to more than twice as many LTFE (205) being known as in Körschens' most extensive study (97). In addition, the setup of new LTFE with a planned duration of at least 20 years goes on and we have also recorded LTFE that were setup after Körschen's publications. In addition, we included grassland experiments.
Also regarding the details to each of the |

| | |
|---|---|
| corresponding justification. This should also explain why the work of Koerschens et al. is obviously not adequate to follow the objectives of this study. | experiments we provide much more information in our dataset (http://doi.org/10.20387/BonaRes-3tr6-mg8r). Although most of the details are not needed for the spatial analyses of this paper, the precise coordinates of the LTFE are needed and could only be found out through our extensive search. |
| Lines 63-80: after the objectives of the work have been formulated in lines 61-63, the explanations given here seem like a description of material and methods. I recommend to shorten this part and to integrate it into the chapter Material and Methods. | We shortened it and enhanced the structure. Parts were integrated into Material and Methods, other parts in the results section. |
| Line 68: what is meant by research parameters? Please list. | By research parameter we mean everything that has ever been sampled and recorded in LTFE. Probably "measured parameters" is less misunderstanding. We changed that. An overview of the measured parameters known to us can be found on pages 9 to 11 of the fact sheet (Grosse, M., Heinrich, U., and Hierold, W.: Fact Sheet for the Description of Long-Term Field Experiments / Steckbrief zur Beschreibung von Dauerfeldversuchen, http://doi.org/10.20387/BonaRes-R56G-FGRW, 2019.). We referred to that. |
| Line 95: here, too, the technical justification for the selected research topics is missing. Especially with regard to the aspect of a meta-analysis of the research statements, which was prominently emphasized in the introduction, the research topics listed here appear incomplete. | We added "descriptive" and skipped "experimental design", which probably lead to the misunderstanding. |
| Lines 200-206: the description of the methodology belongs in the corresponding chapter and is superfluous here, as are lines 208 and 209. Similar mixtures of results and material and methods are also shown in the following chapters. I would recommend to check the results part and to concentrate all methodical information at the appropriate place. | We skipped lines 200 to 206 and enhanced the results part. |
| Figure 1 does not seem necessary to me, the content is very simple and directly repeats the statements in the text without a gain in information. | We would like to leave Figure 1, because we believe it improves the readability of the paper. |
| The core statements in figure 3 could certainly be presented much more clearly. At the moment most of the space is taken up by the legend. It also seems unusual to me that the figure itself contains a headline ('Start of LTFE'). | We enhance Figure 3 and skipped the headline. |

---

## Author Response (AR2)

Dear Prof. Dr. Fiener,

Thank you for the message and the positive feedback. My co-authors and I are delighted that the paper can be published in SOIL.

I made corrections according to the identified needs, which are indicated in the following manuscript by tracked-changes:

Lines 52-54: I included a short explanation about "Bodendauerbeobachtungsflächen"

Line 72: I deleted one sentence, because the source is not published yet

Lines 282-288 (Chapter 3.2.1): I explained the German terms

Lines 345-346: I stated "C-model" more clearly

References: I made several corrections and deleted one source which is not published yet

Figure 3: I changed the small circles into a line

Thank you very much for your work and support as Topical Editor.

Best regards,

Meike Grosse

[revised manuscript text omitted]